# $q$-EXPONENTIAL FAMILY FOR POLICY OPTIMIZATION

**Lingwei Zhu**[*]
University of Tokyo
lingwei4@ualberta.ca

**Haseeb Shah**[*]
University of Alberta
hshah1@ualberta.ca

**Han Wang**[*]
University of Alberta
han8@ualberta.ca

**Yukie Nagai**
University of Tokyo

**Martha White**
University of Alberta

## ABSTRACT

Policy optimization methods benefit from a simple and tractable policy parametrization, usually the Gaussian for continuous action spaces. In this paper, we consider a broader policy family that remains tractable: the $q$-exponential family. This family of policies is flexible, allowing the specification of both heavy-tailed policies ($q > 1$) and light-tailed policies ($q < 1$). This paper examines the interplay between $q$-exponential policies for several actor-critic algorithms conducted on both online and offline problems. We find that heavy-tailed policies are more effective in general and can consistently improve on Gaussian. In particular, we find the Student's t-distribution to be more stable than the Gaussian across settings and that a heavy-tailed $q$-Gaussian for Tsallis Advantage Weighted Actor-Critic consistently performs well in offline benchmark problems. In summary, we find that the Student's t policy a strong candidate for drop-in replacement to the Gaussian. Our code is available at https://github.com/lingweizhu/qexp.

## 1 INTRODUCTION

Policy optimization methods optimize the parameters of a stochastic policy towards maximizing some performance measure (Sutton et al., 1999). These methods benefit from a simple and tractable policy functional. For discrete action spaces, the Boltzmann-Gibbs (BG) policy is often preferred (Mei et al., 2020; Cen et al., 2022); while the Gaussian policy is standard for the continuous case. For continuous action spaces, sampling the BG policy is computationally expensive due to the normalizing log-partition function. A Gaussian policy is often used as a tractable approximation. While there are other candidates such as the Beta policy (Chou et al., 2017), the Gaussian remains the most common choice for both online and offline policy optimization methods (Haarnoja et al., 2018; Neumann et al., 2023; Xiao et al., 2023).

In this paper, we consider a broader policy family that remains tractable called the $q$-exponential family. The $q$-exponential family was proposed to study non-extensive system behaviors in the statistical physics (Naudts, 2010; Matsuzoe & Ohara, 2011), and has recently been exploited in transformers (Peters et al., 2019; Martins et al., 2022). By setting $q = 1$, it recovers the standard exponential family. With $q > 1$, we can obtain policies with heavier tails than the Gaussian, such as the Student's t-distribution (Kobayashi, 2019) or the Lévy Process distribution (Simsekli et al., 2019; Bedi et al., 2024). Heavy-tailed distributions can preferable as they are more robust (Lange et al., 1989), can facilitate exploration and help escape local optima in the sparse reward context (Chakraborty et al., 2023). When $q < 1$, light-tailed (sparse) policies such as the $q$-Gaussian distribution can be recovered. The sparse $q$-Gaussian has finite support and

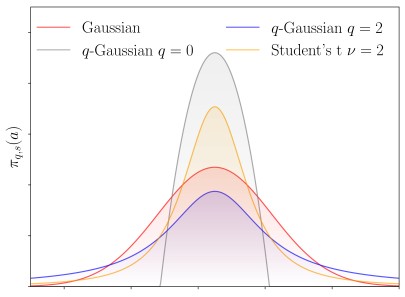

Figure 1: The policy parametrizations in this paper. Student's t is a $q$-exponential with $q = 1 + 2/(\nu + 1) \approx 1.67$.

---

[*]indicates joint first authors.

| Reference | Scope of $\exp_q$ | Explicit? | Heavy & Sparse? | Continuous? | RL? |
|---|---|---|---|---|---|
| Naudts (2010); Matsuzoe & Ohara (2011) | $q \in \mathbb{R}$ | - | ✓ | ✓ | ✗ |
| Martins et al. (2022) | $q < 1$ | - | ✗ | ✓ | ✗ |
| Lee et al. (2018); Chow et al. (2018a) | $q = 0$ | ✓ | ✗ | ✗ | ✓ |
| Lee et al. (2020); Zhu et al. (2023; 2024) | $q < 1$ | ✗ | ✗ | ✓ | ✓ |
| Li et al. (2023) | $q = 0$ | ✓ | ✗ | ✓ | ✓ |
| This paper | $q \in \mathbb{R}$ | ✓ | ✓ | ✓ | ✓ |

Table 1: Existing works and their scopes. We are the first to consider the general $q$-exponential family for the parameterized policy in reinforcement learning. The family includes continuous heavy-tailed and sparse policies. Prior works in RL considered only the discrete case or continuous policy with a specific entropic index $q$. Further, in many cases they still used a Gaussian policy parameterization to approximate an implicit target distribution that is a $q$-exponential, rather than explicitly using the $q$-Gaussian as the policy parameterization.

can serve as a continuous generalization of the discrete sparsemax. As a result, $q$-Gaussian helps alleviate safety concerns incurred by the infinite support Gaussian (Xu et al., 2023; Li et al., 2023).

Such $q$-exponential families have been considered in reinforcement learning, with the existing work summarized in Table 1. Lee et al. (2018); Chow et al. (2018b) studied the discrete setting with $q = 0$, called the sparsemax. Li et al. (2023) similarly considered $q = 0$ policy parameterization for the continuous action setting. All other works, however, used a Gaussian policy parameterization to (implicitly) approximate an idealized target distribution that is $q$-Gaussian, and specifically for $q < 1$ Lee et al. (2020); Zhu et al. (2024). Such a choice is suboptimal, as Gaussians are used to approximate light-tailed (sparse) target policies. And in fact that choice was not strictly necessary as the policy parameterization need not have been chosen to be Gaussian: it could also have been a $q$-Gaussian. Though obvious in hindsight, this gap was likely due to simply not considering the use of general continuous $q$-exponential family for the policy parameterization, which is what we introduce in this work.

In this paper, we empirically investigate the $q$-exponential family as a replacement for the Gaussian inside several existing policy optimization algorithms. Our contributions include the following. (1) We show how to use $q$-exponential family policy parameterizations inside a variety of existing actor-critic algorithms. (2) We provide comprehensive experiments on both online and offline problems showing that $q$-exponential family policies can improve on the Gaussian by a large margin. In particular, we find that the Student's t policy is more stable, performing well across algorithms and problems, shown in Figure 2. (3) We provide empirical evidence supporting the assumption that algorithms may prefer specific policies depending on the actor loss objective. In particular, we find that by replacing the Gaussian with a heavy-tailed $q$-Gaussian, Tsallis Advantage Weighted Actor-Critic (Zhu et al., 2024) consistently performs better across offline benchmark problems. This outcome makes sense; as mentioned above, this algorithm implicitly has a target policy that is a $q$-Gaussian, so using a matching $q$-Gaussian parameterization should perform better.

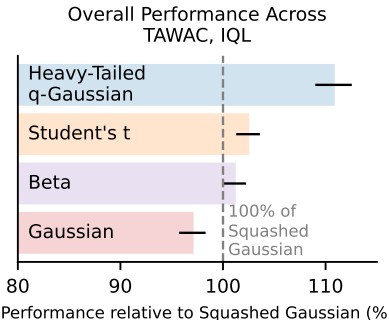

Figure 2: Performance relative to the Squashed Gaussian on the offline D4RL MuJoCo task, averaged across the selected algorithms and environments.

## 2 BACKGROUND

We focus on discounted Markov Decision Processes (MDPs) expressed by the tuple $(\mathcal{S}, \mathcal{A}, P, \mu, r, \gamma)$, where $\mathcal{S}$ and $\mathcal{A}$ denote state space and action space, respectively. Let $\Delta(\mathcal{X})$ denote the set of probability distributions over $\mathcal{X}$. $P$ and $\mu$ denote the transition probability and initial state distribution, respectively. $r(s, a)$ defines the reward associated with that transition. $\gamma \in (0, 1)$ is the discount factor.

A policy $\pi : \mathcal{S} \to \Delta(\mathcal{A})$ is a mapping from the state space to distributions over actions. To assess the quality of a policy, we define the expected return as $J(\pi) = \int_{\mathcal{S}} \rho^\pi(s) \int_{\mathcal{A}} \pi(a|s) r(s, a) \mathrm{d}a \mathrm{d}s$, where $\rho^\pi(s) = \sum_{t=0}^{\infty} \gamma^t P(s_t = s)$ is the unnormalized state visitation frequency. The goal is to learn a policy that maximizes $J(\pi)$. We also define the action value and state value as $Q^\pi(s, a) = \mathbb{E}_\pi \left[ \sum_{t=0}^{\infty} \gamma^t r(s_t, a_t) | s_0 \sim \mu, a_0 = a \right]$, $V^\pi(s) = \mathbb{E}_\pi \left[ Q^\pi(s, a) \right]$. For the ease of later notations, we write the dependence on state as subscript, e.g. $Q(s, a)$ will be written as $Q_s(a)$.

In practice, the policy is often parametrized by a vector of parameters $\theta \in \mathbb{R}^n$. The policy can then be optimized by adjusting its parameters to the high reward region utilizing its gradient information.

The Policy Gradient Theorem (Sutton et al., 1999) featured by many policy gradient methods states that the gradient can be computed by:

$$\nabla_\theta J(\pi; \theta) = \mathbb{E}_{s \sim \rho^\pi, a \sim \pi_\theta} \left[ Q_s^\pi(a) \nabla_\theta \ln \pi_s(a; \theta) \right].$$

In practice, the expectation is approximated by sampling. When the state space is large, the action value function is also parametrized, leading to the Actor-Critic methods (Degris et al., 2012).

In contrast to the study of policy gradient algorithms, the impact of specific policy parametrizations on performance remains a less studied topic. Researchers typically consider policy parametrizations that can be written as the following:

$$\pi_s(a; \theta) = \frac{1}{Z_s} \exp \left( \theta^\top \phi_s(a) \right) = \exp \left( \theta^\top \phi_s(a) - Z_s' \right). \tag{1}$$

Here, $\phi_s(a)$ is a vector of statistics and $\theta \in \mathbb{R}^n$ is a vector of parameters, $Z_s$ is the normalizing constant ensuring the policy is a valid distribution and $Z_s := \exp \left( Z_s' \right)$. One immediate instance is the Boltzmann-Gibbs (BG) policy $\pi_{\mathrm{BG}, s}(a; \theta) = \exp \left( Q_s(a) - Z_s \right)$, where $Z_s' = \ln \int \exp \left( Q_s(a) \right) \mathrm{d}a$ is the log-partition function. In the discrete case, it is also called the *softmax transformation* (Cover & Thomas, 2006). BG policy has been studied extensively in RL for encouraging exploration and smoothing the optimization landscape, to name a few applications (Haarnoja et al., 2018; Ahmed et al., 2019; Cen et al., 2022). However, evaluating the log-partition function is in general intractable.

## 3 EXPONENTIAL AND $q$-EXPONENTIAL FAMILIES

We first review the commonly used policy parametrizations. They permit an expression using the exponential function. We arrive at the more general $q$-exponential family by deforming the exponential. In Table 2, we summarize all policies presented in the paper.

### 3.1 THE EXPONENTIAL FAMILY POLICIES

The Gaussian policy is one of the simplest distributions one can consider due to its omnipresence in statistics and parametric estimation as well as its widely available sampling procedure implementations. Since evaluating the log-partition function of BG is intractable, due to the aforementioned advantages many researchers consider the Gaussian policy instead: $\pi_s(a) = \frac{1}{\sqrt{2\pi}\sigma_s} \exp \left( \frac{-(a - \mu_s)^2}{2\sigma_s^2} \right)$. For simplicity we drop the dependence on state $s$. To see it is a member of the exponential family, in Eq. (1) let $\theta = [\frac{\mu}{\sigma^2}, -\frac{1}{2\sigma^2}]^\top$ for $\mu \in (-\infty, \infty), \sigma > 0$; $\phi_s(a) = [a, a^2]^\top$, and $Z_s = \ln \left( \sqrt{2\pi}\sigma \right)$. This amounts to setting $Q_s(a) = -\frac{(a - \mu)^2}{2\sigma^2}$ in the BG (Gu et al., 2016). We write a Gaussian policy as $\pi_{\mathcal{N}, s}(a) = \mathcal{N}(a; \mu, \sigma^2)$. The gradients of the Gaussian are $\nabla_\mu \ln \pi_s(a) = \frac{(a - \mu)}{\sigma^2}$ and $\nabla_\sigma \ln \pi_s(a) = \frac{(a - \mu)^2}{\sigma^3} - \frac{1}{\sigma}$. On one hand, the Gaussian policy is simple to implement. On the other hand, when $\sigma$ becomes small, Gaussian can be unstable due to overly large gradients and can prematurely concentrate on a suboptimal action. As a result, it is susceptible to noise/outliers and does not encourage sufficient exploration due to its thin tails. This paper investigates location-scale alternatives within the generalized $q$-exponential family.

Another interesting member is the Beta distribution (Chou et al., 2017): $\pi_{\mathrm{Beta}, s}(a) = \frac{\Gamma(\alpha + \beta)}{\Gamma(\alpha)\Gamma(\beta)} a^{\alpha - 1} (1 - a)^{\beta - 1}, \ a \in (0, 1)$, where $\Gamma(\cdot)$ is the gamma function. It can be retrieved from equation 1 by letting $\theta = [\alpha, \beta]^\top, \phi_s(a) = [\ln a, \ln(1 - a)]^\top, Z_s = \frac{\Gamma(\alpha)\Gamma(\beta)}{\Gamma(\alpha + \beta)}$. Since Beta distribution's support is bounded between $(0, 1)$, Chou et al. (2017) argued that it might alleviate the

| Family | Policy | Parameters $\theta$ | Statistics $\phi_s(a)$ | Normalization $Z_s$ | $\nabla \ln \pi_s(a)$ |
|---|---|---|---|---|---|
| exp | Gaussian | $\left[\frac{\mu}{\sigma^2}, -\frac{1}{2\sigma^2}\right]$ | $[a, a^2]$ | $\sqrt{2\pi}\sigma$ | Eq. (13) |
| | Beta | $[\alpha, \beta]$ | $[\ln a, \ln(1-a)]$ | $\frac{\Gamma(\alpha)\Gamma(\beta)}{\Gamma(\alpha+\beta)}$ | - |
| $q$-exp | Student's t | $\left[\frac{-2\mu}{\nu\sigma}, \frac{1}{\nu\sigma}\right]$ | $[a, a^2]$ | $\frac{\sqrt{\pi\nu\sigma}\,\Gamma\left(\frac{\nu}{2}\right)}{\Gamma\left(\frac{\nu+1}{2}\right)}$ | Eq. (14) |
| | $q$-Gaussian ($q < 1$) | $\left[\frac{\mu}{\sigma^2}, -\frac{1}{2\sigma^2}\right]$ | $[a, a^2]$ | $\sqrt{\frac{\pi}{1-q}}\frac{\Gamma\left(\frac{1}{1-q}+1\right)}{\Gamma\left(\frac{1}{1-q}+\frac{3}{2}\right)}$ | Eq. (15) |
| | $q$-Gaussian ($1 < q < 3$) | | | $\sqrt{\frac{\pi}{q-1}}\frac{\Gamma\left(\frac{1}{q-1}-\frac{1}{2}\right)}{\Gamma\left(\frac{1}{q-1}\right)}$ | |

Table 2: Policy parametrizations from the exp and $q$-exp families studied in this paper. We are primarily interested in the location-scale family. Their multivariate forms are shown in Appendix A.

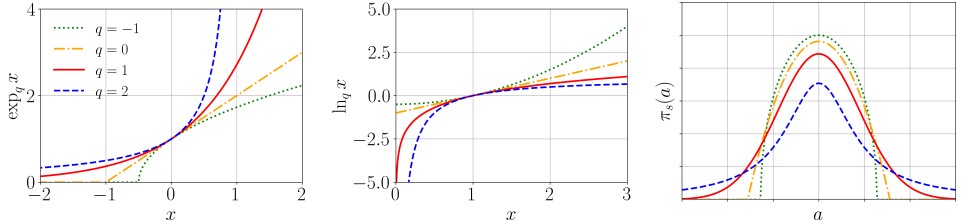

Figure 3: $\exp_q x$ and $\ln_q x$ for $q < 1$ and $q > 1$. When $q = 1$ they respectively recover their standard counterpart. For $q < 1$ the $q$-exp can return zero values and hence $q$-exp policies may achieve sparsity. For $q > 1$, $q$-exp decays more slowly towards 0, resulting in heavy-tailed behaviors. The rightmost shows the $q$-Gaussian with different $q$.

bias introduced by truncating Gaussian densities outside the action space bounds. The beta policy is the only non-location-scale family distribution in this paper. However, as we will show in the experiments, the Beta policy generally does not perform favourably against the Gaussian.

## 3.2 THE $q$-EXPONENTIAL FAMILY, HEAVY-TAILED AND LIGHT-TAILED POLICIES

Generalizing the exponential family using the $q$-exponential function has been extensively discussed in statistical physics (Naudts, 2002; Tsallis, 2009; Naudts, 2010; Amari & Ohara, 2011). In the machine learning literature, the $q$-exponential generalization has attracted some attention since it allows for tuning the tail behavior by adjusting the value of $q$ (Sears, 2008; Ding & Vishwanathan, 2010; Amid et al., 2019). The $q$-exponential and its unique inverse function $q$-logarithm are:

$$\exp_q x := \begin{cases} \exp x, & q = 1 \\ [1 + (1-q)x]_+^{\frac{1}{1-q}}, & q \neq 1 \end{cases}, \quad \ln_q x := \begin{cases} \ln x, & q = 1 \\ \frac{x^{1-q}-1}{1-q}, & q \neq 1 \end{cases} \quad (2)$$

where $[\cdot]_+ := \max\{\cdot, 0\}$. $q$-exp/log generalize exp/log since $\lim_{q \to 1} \exp_q x = \exp x$ and $\lim_{q \to 1} \ln_q x = \ln x$. Similar to exp, $q$-exp is an increasing and convex function for $q > 0$, satisfying $\exp_q(0) = 1$. However, an important difference of $q$-exp is that $\exp_q(a + b) \neq \exp_q(a)\exp_q(b)$ unless $q = 1$. We visualize $q$-exp/log in Figure 3.

We now define the $q$-exponential family as:

$$\pi_{q,s}(a; \theta) = \frac{1}{Z_{q,s}} \exp_q\left(\theta^\top \phi_s(a)\right) = \exp_q\left(\theta^\top \phi_s(a) - Z'_{q,s}\right), \quad (3)$$

where $\theta, \phi_s(a), Z_{q,s}$ have similar meanings to equation 1. Note that $Z_{q,s} \neq \exp_q\left(Z'_{q,s}\right)$ unless $q = 1$. The $q$-exponential family includes the $q$-Gaussian and Student's t distributions described in the next subsections.

### 3.2.1 $q$-GAUSSIAN

As the counterpart of Gaussian in the $q$-exp family, $q$-Gaussian unifies both light-tailed and heavy-tailed policies by varying the entropic index $q$ (Matsuzoe & Ohara, 2011):

$$\pi_{\mathcal{N}_q,s}(a) = \frac{1}{Z_{q,s}} \exp_q\left(-\frac{(a-\mu)^2}{2\sigma^2}\right),$$

$$\text{where } Z_{q,s} = \begin{cases} \sigma\sqrt{\frac{\pi}{1-q}}\, \Gamma\left(\frac{1}{1-q}+1\right) \big/ \, \Gamma\left(\frac{1}{1-q}+\frac{3}{2}\right) & \text{if } -\infty < q < 1, \\ \sigma\sqrt{\frac{\pi}{q-1}}\, \Gamma\left(\frac{1}{q-1}-\frac{1}{2}\right) \big/ \, \Gamma\left(\frac{1}{q-1}\right) & \text{if } 1 < q < 3. \end{cases} \tag{4}$$

It is heavy-tailed when $1 < q < 3$ and light-tailed when $q < 1$. $\pi_{\mathcal{N}_q,s}(a)$ is no longer integrable for $q \geq 3$ (Naudts, 2010). We visualize these $q$-Gaussians in Figure 3.

Since popular libraries like the PyTorch (Paszke et al., 2019) do not have implementations of $q$-Gaussians available, we discuss their sampling methods. It was shown by (Martins et al., 2022) that a sparse $q$-Gaussian ($q < 1$) random variable permits a stochastic representation $\boldsymbol{\mu} + rA\boldsymbol{u}$, where $\boldsymbol{u} \sim \texttt{Unif}\left(\mathbb{S}^N\right)$ is a random sample from the $N-1$ dimensional unit sphere. $A$ is the scaled matrix $|\Sigma|^{-\frac{1}{2N+\frac{4}{1-q}}}\Sigma^{\frac{1}{2}}$. $r$ is the radius of the distribution, and the ratio follows the Beta distribution $r^2/R^2 \sim \texttt{Beta}\left((2-q)/(1-q), N/2\right)$, where $R$ is radius of the supporting sphere of the standard $q$-Gaussian $\mathcal{N}_q(0, I)$:

$$R = \left(\frac{\Gamma\left(\frac{N}{2} + \frac{2-q}{1-q}\right)}{\Gamma\left(\frac{2-q}{1-q}\right)\pi^{\frac{N}{2}}} \cdot \left(\frac{2}{1-q}\right)^{\frac{1}{1-q}}\right)^{\frac{1-q}{2+(1-q)N}}. \tag{5}$$

Notice that $R$ depends only on the dimensionality $N$ and the entropic index $q$. This method provides low-variance samples, but unfortunately it does not extend to $q > 1$. Therefore, for $1 < q < 3$ we adopt the Generalized Box-Müller Method (GBMM) (Thistleton et al., 2007) to transform uniform random variables $\boldsymbol{u}_1, \boldsymbol{u}_2 \sim \texttt{Unif}(0,1)^N$ by the following:

$$\boldsymbol{z}_1 = \sqrt{-2\ln_q(\boldsymbol{u}_1)} \cdot \cos(2\pi\boldsymbol{u}_2), \qquad \boldsymbol{z}_2 = \sqrt{-2\ln_q(\boldsymbol{u}_1)} \cdot \sin(2\pi\boldsymbol{u}_2). \tag{6}$$

Then each of $\boldsymbol{z}_1, \boldsymbol{z}_2$ is a standard $q$-Gaussian variable with new entropic index $q' = (3q-1)/(q+1)$. Often we know the desired $q'$ in advance, in this case we simply let the $q$-log take on the index $q = (q'-1)/(3-q')$. The desired random vector is given by $\boldsymbol{\mu} + \Sigma^{\frac{1}{2}}\boldsymbol{z}$.

### 3.2.2 STUDENT'S T

Heavy-tailed distributions like the Student's t are popular for robust modelling (Lange et al., 1989). The Student's t distribution is

$$\pi_{\text{St},s}(a) = \frac{\Gamma\left(\frac{\nu+1}{2}\right)}{\sqrt{\pi\nu}\sigma\,\Gamma\left(\frac{\nu}{2}\right)} \left(1 + \frac{(a-\mu)^2}{\sigma\nu}\right)^{-\frac{\nu+1}{2}}, \tag{7}$$

where $\nu > 0$ is the degree of freedom. As $\nu \to \infty$, Student's t distribution approaches the Gaussian. Numerically, Student's t with $\nu \geq 30$ is considered to closely match the Gaussian. Therefore, $\nu$ can be an important learnable parameter in addition to its location $\mu$ and scale $\sigma$. It allows the policy to adaptively balance the exploration-exploitation trade-off by interpolating the Gaussian and heavy-tailed policies. Now let $q = 1 + \frac{2}{\nu+1}$ and define

---

**Algorithm 1:** $q$-Gaussian sampling

**Input:** $q, N, \mu, \Sigma$
**if** $q < 1$ **then**
    sample $\boldsymbol{u} \sim \texttt{Unif}(\mathbb{S}^N)$
    sample $z \sim \texttt{Beta}\left(\frac{2-q}{1-q}, \frac{N}{2}\right)$
    compute $R$ per Eq. (5)
    compute $A = |\Sigma|^{-\frac{1}{2N+\frac{4}{1-q}}}\Sigma^{\frac{1}{2}}$
    **return** $\boldsymbol{\mu} + \sqrt{zR^2}A\boldsymbol{u}$
**else if** $q > 1$ **then**
    sample $\boldsymbol{u}_1, \boldsymbol{u}_2 \sim \texttt{Unif}(0,1)^N$
    compute $\boldsymbol{z}$ by GBMM Eq. (6)
    **return** $\boldsymbol{\mu} + \Sigma^{\frac{1}{2}}\boldsymbol{z}$

---

$$Z_{q,s} := \frac{\sqrt{\pi\nu}\sigma\,\Gamma\left(\frac{\nu}{2}\right)}{\Gamma\left(\frac{\nu+1}{2}\right)}, \quad \theta^\top\phi_s(a) := \frac{Z_{q,s}^{q-1}}{(1-q)}\frac{(a-\mu)^2}{\sigma\nu},$$

$$\Rightarrow \quad \pi_{\text{St},s}(a) = \exp_q\left(\theta^\top\phi_s(a) - \ln_{2-q}Z_{q,s}\right), \tag{8}$$

which we see it is indeed a $q$-exp policy and $Z'_{q,s} = \ln_{2-q} Z_{q,s}$. Student's t policy has been used in (Kobayashi, 2019) to encourage exploration and to escape local optima. Another related case is the Cauchy's distribution recovered when $q = 2$ (or $\nu = 1$ from Student's t). Cauchy's distribution can be used as the starting point for learning Student's t. Note that Cauchy's distribution does not have valid mean, variance or any higher moments.

## 4  USING $q$-EXPONENTIAL FAMILIES FOR ACTOR-CRITIC ALGORITHMS

In this section, we outline three key actor-critic algorithms we use in our study and the nuances of incorporating $q$-exp policies into them. For example, the $q$-exp policies may not have closed-form Shannon entropy. Therefore, approximations are needed for algorithms like SAC and GreedyAC. Moreover, though for the Gaussian evaluating the log-likelihood for off-policy/offline actions causes no problem, it raises a new issue for the light-tailed $q$-Gaussian, since these actions can fall outside of its support.

**Soft Actor-Critic.**  SAC (Haarnoja et al., 2018) encourages exploration by adding to reward the Shannon entropy. The actor minimizes the following KL loss

$$\mathcal{L}_{\text{SAC}}(\phi) := \mathbb{E}_{s \sim \mathcal{B}}\left[D_{KL}(\pi_{\phi,s} \,\|\, \pi_{\text{BG},s})\right] = \mathbb{E}_{s \sim \mathcal{B}}\left[D_{KL}\left(\pi_{\phi,s} \,\left\|\, \frac{\exp\left(\tau^{-1}Q_s\right)}{Z_s}\right.\right)\right],$$

where states are sampled from replay buffer $\mathcal{B}$. The parametrized policy $\pi_\phi$ is projected to be close to the BG policy. By default $\pi_\phi$ is chosen to be the Gaussian policy, but potentially a more exploring policy like the Student's t could be better. Depending on action values, BG can have multiple modes and heavy tails. The Gaussian may not be able to fully capture these characteristics.

**Greedy Actor-Critic.**  GreedyAC (Neumann et al., 2023) maintains an additional proposal policy for exploration by maximizing Shannon entropy augmented rewards. Its actor policy maximizes unbiased reward and learns from the high-quality actions generated by the proposal policy. To simplify notations, we use $I(s)$ to denote the set of high quality actions given $s$.

$$\mathcal{L}_{\text{GreedyAC, prop}}(\phi) := \mathbb{E}_{\substack{s \sim \mathcal{B} \\ a \in I(s)}}\left[-\ln \pi_{\phi,s} - \mathcal{H}\left(\pi_{\phi,s}\right)\right],$$

$$\mathcal{L}_{\text{GreedyAC, actor}}(\bar{\phi}) := \mathbb{E}_{\substack{s \sim \mathcal{B} \\ a \in I(s)}}\left[-\ln \pi_{\bar{\phi},s}\right].$$

GreedyAC maximizes log-likelihood of the actor and entropy-augmented likelihood for the proposal policy. Note that the when $\pi_{\phi,s}$ is a $q$-exp policy, it may not have a closed-form Shannon entropy expression. Therefore, we can use log-probabilities as a surrogate just like in SAC.

**Tsallis Advantage Weighted Actor-Critic.**  TAWAC (Zhu et al., 2024) proposed to use a light-tailed $q$-exp policy for offline learning. However, the light-tailed distribution was approximated with the Gaussian which is an infinite-support policy. Let $\pi_{\mathcal{D}}$ denote the empirical behavior policy and $\mathcal{D}$ the offline dataset. TAWAC minimizes the following actor loss:

$$\mathcal{L}(\phi) := \mathbb{E}_{s \sim \mathcal{D}}\left[D_{KL}(\pi_{\text{TKL},s} \,\|\, \pi_{\phi,s})\right] = \mathbb{E}_{\substack{s \sim \mathcal{D} \\ a \sim \pi_{\mathcal{D}}}}\left[-\exp_{q'}\left(\frac{Q_s(a) - V_s}{\tau}\right)\ln \pi_{\phi,s}(a)\right], \quad (9)$$

where $\pi_{\text{TKL},s}(a) \propto \pi_{\mathcal{D},s}(a)\exp_{q'}\left(\tau^{-1}\left(Q_s(a) - V_s\right)\right)$ denotes the Tsallis KL regularized policy. $\pi_{\phi,s}$ mimics a TKL policy which can be sparse depending on $q'$. In this case, it is natural to expect that a sparse policy parametrization may lead to better performance.

Algorithms like TAWAC that sample from a behavior policy $\pi_{\mathcal{D}}$ needs extra caution when using the $q$-exp policies. When the light-tailed $q$-Gaussian is used, numerical issues can be incurred since the action sampled may fall outside the support of $\pi_\phi$, leading to undefined log-likelihood. To resolve this problem, we propose to sample from $\pi_\phi$ a batch of $K$ actions and replace the out-of-support action with the in-support one with least $L_2$ distance, see Alg. 2.

---

**Algorithm 2:** Out-of-support action handling for the light-tailed $q$-Gaussian

**Input:** out-of-support action $a$
sample in-support actions $\{\boldsymbol{b}_i\}_{i=1:K}$
solve $i^* = \arg\min_i \|\boldsymbol{b}_i - \boldsymbol{a}\|_2^2$
**return** $\boldsymbol{b}_{i^*}$

---

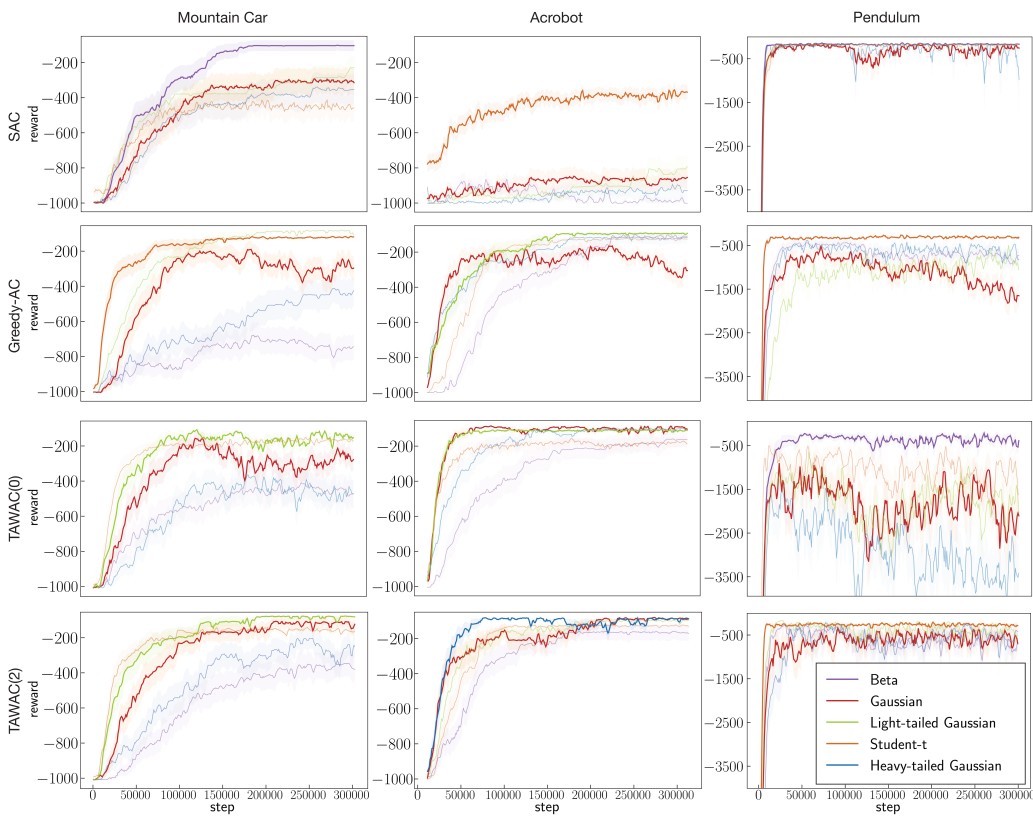

Figure 4: Learning curves on the classic control environments. Only the Gaussian and the best policy parametrization for each setting were shown with full opacity. The best policy is picked based on the total area under the curve (AUC). TAWAC(0) refers to TAWAC with entropic index $q' = 0$ in Eq. (9). Despite tuning hyperparameters separately for each policy, Gaussian is the best policy in only $1/12$ settings. In most other settings, the Gaussian policy performs significantly worse than the best.

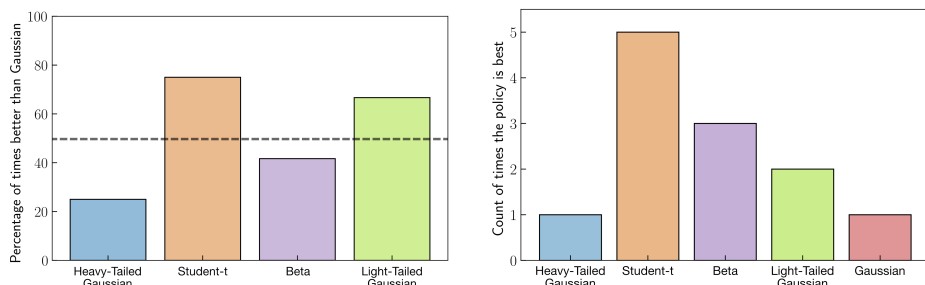

Figure 5: (Left) The percentage of times that each policy parametrization is better than the Gaussian across all algorithm-environment combinations based on total AUC. If the bar is above the $50\%$ line, then it means that the policy parametrization is better than Gaussian on average. We see that Student's t and Light-tailed Gaussians are better than the Gaussian in $75\%$ and $66\%$ of the settings, respectively. (Right) Count of times where a policy parametrization performed the best across all algorithm-environment combinations based on AUC. We observe that the student-t policy performed the best in $5/12$ settings, whereas the Gaussian policy performed the best only once.

## 5 EXPERIMENTS

Our empirical study's primary goal is to understand better the performance differences under this broader class of policy parameterizations in both online and offline settings. We ran experiments

with different algorithms, to get a better sense of how conclusions about policy parameterization vary across different actor-critic algorithms.

We parametrize Student-t's DOF parameter $\nu$ in addition to its location and scale. By contrast, the heavy-tailed $q$-Gaussian is fixed at $q = 2$, since its allowable range is $1 < q < 3$. For the light-tailed $q$-Gaussian, we opt for the standard choice of $q = 0$. Since Student's t, heavy-tailed q-Gaussian, and Gaussian have unbounded support, we clipped the sampled action to fit the task's action space without modifying the density. We swept the hyperparameters using five random seeds, then increased the number of seeds to 10 for the best parameter setting. The hyperparameter sweep ranges and the best values are provided in Appendix D.2 and D.3.

## 5.1 ONLINE CLASSIC CONTROL

**Domains and Baselines.**   We used three classical control environments in the continuous action setting: Mountain Car (Sutton & Barto, 2018), Pendulum (Degris et al., 2012) and Acrobot (Sutton & Barto, 2018). We chose the cost-to-goal version of Mountain Car, which outputs $-1$ reward per time step to encourage reaching the goal early. We compared SAC, GreedyAC and two versions of TAWAC, $q' = 0$ and $q' = 2$.

**Results.**   Figure 4 shows the learning curves of all algorithm-environment combinations. Only the Gaussian and the environment-specific best policy are shown with full opacity, computed based on area under curve (AUC). One immediate observation is that, though all three algorithms by default choose the Gaussian policy, it was seldom the best policy parametrization. Environment-wise, on Mountain Car the Gaussian did not rank the best for any of the algorithms. By contrast, the Beta policy attained the first place with SAC, as was the light-tailed $q$-Gaussian with TAWAC. The same trend for the Gaussian holds in Acrobot and Pendulum as well, with exception only on TAWAC(0) Acrobot, where its curve closely resembled that of the light-tailed $q$-Gaussian.

Algorithm-wise, three observations are to be made: (i) on Mountain Car the Beta policy performed significantly better than others. This could be due to its flexibility in maintaining a skewed distribution shape that matches the BG policy more closely in contrast to the other location scale family members. (ii) The $q$-Gaussians in general outperformed the Gaussian on TAWAC(0) and TAWAC(2) whose actor explicitly mimics a $q$-exp policy. (iii) Student's t has ranked the top involving all three algorithms. Figure 5 LHS summarizes the percentage of each policy parametrization outperforming the Gaussian. The Student's t and light-tailed Gaussian went above 50%, suggesting potentially greater applicability. The RHS shows out of 12 total combinations, how many times each policy parametrization has ranked the top. The result shows that the Student's t attained 5 times, contrasting the 1 time of the Gaussian.

In Figure 6 we visualized the evolution of Gaussian and $q$-Gaussian policies on the starting state over the first $4 \times 10^4$ steps (10% of the entire learning horizon). Note that the allowed action range is $[-1, 1]$ but the plot shows $[-2, 2]$ for better visualization. Gaussian tends to quickly concentrate like a delta policy. This can be detrimental to algorithms like SAC and GreedyAC which demand stochasticity to generate diverse samples. By contrast, both light- and heavy-tailed $q$-Gaussians tend to be more stochastic.

In Figure 11 we show the Manhattan plot of SAC with all swept hyperparameters on all environments. Though there is no a definitive winner for all cases, it is visible that the Student's t and Gaussian have a similar behavior to hyperparameter changes. Therefore, if we are tackling a problem where the Gaussian works, the Student-t is very likely to work. And judging from Fig. 5, we know that Student's t is 75% more likely to perform better than Gaussian given the same hyperparameter sweeping range.

Figure 6: Policy evolution of GreedyAC on Mountain Car. The Gaussian collapsed into a delta-like policy after only $10\%$ of the learning horizon.

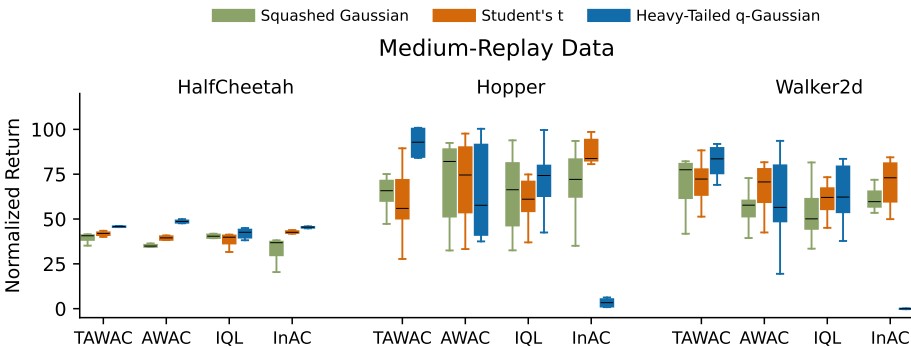

Figure 7: Normalized scores on Medium-Replay level datasets from the MuJoCo suite. The black bar shows the median. Boxes and whiskers are $1\times$ and $1.5\times$ interquartile ranges, respectively. See Figure 15 for full comparison. Environment-wise, TAWAC with heavy-tailed $q$-Gaussian is often the top performer. Algorithm-wise, Student's t consistently outperforms Squashed Gaussian.

## 5.2 OFFLINE D4RL MuJoCo

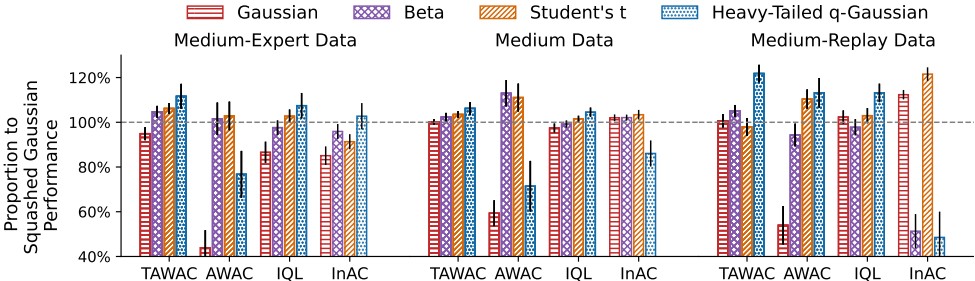

Figure 8: Relative improvement to the Squashed Gaussian policy, averaged over multiple environments in the MuJoCo suite. The Student's t consistently outperforms the Gaussian with all the chosen algorithms. The heavy-tailed $q$-Gaussian with TAWAC and IQL also achieved significant improvement. The improvement can reach up to $\sim 20\%$. Black vertical lines at the top indicate one standard error.

**Domains and Baselines.** We used the standard benchmark MuJoCo suite from D4RL to evaluate algorithm-policy combinations (Fu et al., 2020). The following algorithms are compared: TAWAC, Advantage Weighetd Actor-Critic (AWAC) (Nair et al., 2021), Implicit Q-Learning (IQL) (Kostrikov et al., 2022), In-sample Actor-Critic (InAC) (Xiao et al., 2023). For TAWAC, we fixed its leading $q'$-exp with $q' = 0$. In Appendix C.2 we detailed the compared algorithms. We also included a popular variant of the Gaussian known as the Squashed Gaussian for comparison. Being able to evaluate the offline log-probability is critical to the tested algorithms, we found that light-tailed $q$-Gaussian leads to poor performance even with random online sampling, hence we do not show them here.

**Results.** Figure 7 compared the normalized scores on the Medium-Replay datasets. It can be seen that environment-wise, TAWAC + heavy-tailed $q$-Gaussian was the top performer, and could improve on the Squashed Gaussian by a non-negligible margin. On HalfCheetah, heavy-tailed $q$-Gaussian attained the best score with every algorithm. Algorithm-wise, the heavy-tailed $q$-Gaussian or/and Student's t were better or equivalent to the Squashed Gaussian, except with AWAC on Hopper. Student's t was stable across algorithms, including these with which heavy-tailed $q$-Gaussian performed poorly (e.g., InAC). This demonstrates the value of the learnable DOF parameter that allows it interpolates the Gaussian. In Appendix E we provided comparison on other policies and datasets.

Figure 8 summarized the relative improvement over the Squashed Gaussian across environments. Several observations can be made: (i) though the Squashed Gaussian outperformed the Gaussian in general, it was seldom the best performer. (ii) the Student's t could consistently perform better

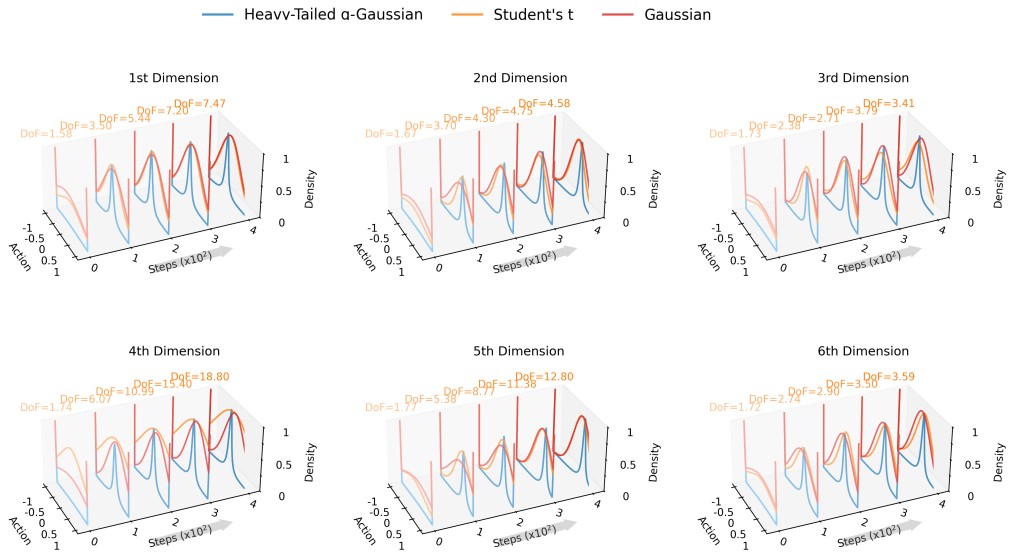

Figure 9: Policy evolution of all actions dimensions of TAWAC on Walker2d Medium Replay. Student's t was flexible in that on some dimensions it had lighter tails like the Gaussian by having large DOF (e.g. 4th), and with heavier tails on the others by having smaller DOF (e.g. 3rd, 6th). The peaks at the edges were caused by clipping actions into the allowed range.

than the Gaussian, the improvement can sometimes reach up to $\sim 20\%$. The same holds for the heavy-tailed $q$-Gaussian with TAWAC and IQL. (iii) though there was no single winner for all cases, choosing the Student's t for the actors with exponential loss functions (AWAC, IQL, InAC), or the heavy-tailed $q$-Gaussian for $q$-exponential actor losses (e.g. TAWAC) are generally effective.

Figure 9 visualized the policy evolution of the Squashed Gaussian and the two heavy-tailed policies, learned with TAWAC on Medium-Replay Walker2D. Squashed Gaussian tended to converge slower here. Since the offline MuJoCo environments are fully deterministic, a wide distribution indicates failure of finding the mode of the optimal action and therefore can be detrimental to learning performance. The Squashed Gaussian converged slower than the heavy-tailed (performed the best) and the Student's t. Student's t was flexible in that it beared lighter tails like the Gaussian in some dimensions by having a large DOF, for example in the 4th and 5th dimensions. On the other hand, it could take heavy tails by having a small DOF like in the 3rd and 6th dimensions.

## 6 CONCLUSION

The Gaussian policy is standard for policy optimization algorithms on continuous action spaces. In this paper we considered a broader family of policies that remains tractable, called the $q$-exponential family. We empirically investigated their utility as a promising alternative to the Gaussian. Specifically, we looked at the Student's t, light- and heavy-tailed $q$-Gaussian policies. Extensive experiments on both online and offline tasks with various actor-critic methods showed that heavy-tailed policies are in general effective. In summary, we found the Student's t policy to be generally more performant and stable than the Gaussian and could be used as a drop-in replacement. By contrast, the Heavy-tailed $q$-Gaussian seemed to favor especially Tsallis regularization and outperformed the baselines.

We acknowledge that the paper has limitations. Perhaps the greatest is the inherent dilemma of the light-tailed $q$-Gaussian evaluating out-of-support actions. Off-policy/offline algorithms require evaluating actions from some behavior policy and the actions can fall outside the support of the sparse $q$-Gaussian. Naïvely discarding these samples results extremely slow or no learning. In this paper we proposed to alleviate this issue by replacing them with the in-support sampled action with the least $L_2$ distance. Nonetheless, this method did not help much in offline experiments. We envision a potential solution that is left to future investigation: projecting the out-of-support actions precisely to the boundary of the $q$-Gaussian.

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

APPENDIX

The Appendix is organized into the following sections. In section A we summarize the multivariate form of $q$-exp policies and derive gradients of their log-likelihood. In section C we discuss the connection between the $q$-exp family and the entropy regularization literature. Based on this, we further discuss how different algorithms may prefer specific policies depending on its actor loss. We then provide implementation details including hyperparameters and how to sample from $q$-Gaussian in section D. Lastly we provide additional experimental results in section E.

## A    MULTIVARIATE DENSITY OF $q$-EXP POLICIES

| Policy | Density | $\nabla \ln \pi_s(a)$ |
|--------|---------|------------------------|
| Gaussian | $\frac{1}{(2\pi)^{\frac{N}{2}} \lvert \Sigma \rvert^{\frac{1}{2}}} \exp\left(-\frac{1}{2}\left(\boldsymbol{a}-\boldsymbol{\mu}\right)^\top \Sigma^{-1}\left(\boldsymbol{a}-\boldsymbol{\mu}\right)\right)$ | Eq. (13) |
| Student's t | $\frac{\Gamma\left(\frac{N+\nu}{2}\right)}{\Gamma\left(\frac{\nu}{2}\right)(\nu\pi)^{\frac{N}{2}} \lvert \Sigma \rvert^{\frac{1}{2}}} \left[1 + \frac{1}{\nu}\left(\boldsymbol{a}-\boldsymbol{\mu}\right)^\top \Sigma^{-1}\left(\boldsymbol{a}-\boldsymbol{\mu}\right)\right]^{-\frac{N+\nu}{2}}$ | Eq. (14) |
| $q$-Gaussian ($q < 1$) | $\frac{(1-q)^{\frac{N}{2}}\Gamma\left(\frac{2-q}{1-q}+\frac{N}{2}\right)}{\Gamma\left(\frac{2-q}{1-q}\right)\pi^{\frac{N}{2}} \lvert \Sigma \rvert^{\frac{1}{2}}} \exp_q\left(-\frac{1}{2}\left(\boldsymbol{a}-\boldsymbol{\mu}\right)^\top \Sigma^{-1}\left(\boldsymbol{a}-\boldsymbol{\mu}\right)\right)$ | Eq. (15) |
| $q$-Gaussian ($1 < q < 3$) | $\frac{(q-1)^{\frac{N}{2}}\Gamma\left(\frac{3-q}{2(q-1)}+\frac{N}{2}\right)}{\Gamma\left(\frac{3-q}{2(q-1)}\right)\pi^{\frac{N}{2}} \lvert \Sigma \rvert^{\frac{1}{2}}} \exp_q\left(-\frac{1}{2}\left(\boldsymbol{a}-\boldsymbol{\mu}\right)^\top \Sigma^{-1}\left(\boldsymbol{a}-\boldsymbol{\mu}\right)\right)$ |  |

Table 3: Multivariate $q$-exp policies and gradients of log-likelihood.

In Table 3 we show multivariate density of the $q$-exp policies introduced in the main text. Note that multivariate Student's t is constructed based on the assumption that a diagonal $\Sigma$ leads to independent action dimensions, same as the Gaussian policy. On the other hand, for $q$-Gaussian this is no longer true, since a diagonal $\Sigma$ does not lead to product of univariate densities.

In the main text we showed their one-dimensional cases for simplicity. For experiments the multivariate densities were used for experiments. We now derive their gradients of log-likelihood with respect to parameters. The following equations will be used frequently (Petersen & Pedersen, 2012):

$$\nabla_{\boldsymbol{\mu}}\left(\boldsymbol{a}-\boldsymbol{\mu}\right)^\top \Sigma^{-1}(\boldsymbol{a}-\boldsymbol{\mu}) = -2\Sigma^{-1}(\boldsymbol{a}-\boldsymbol{\mu}), \tag{10}$$

$$\nabla_\Sigma \ln \lvert \Sigma \rvert = \left(\Sigma^\top\right)^{-1}, \tag{11}$$

$$\nabla_\Sigma(\boldsymbol{a}-\boldsymbol{\mu})^\top \Sigma^{-1}(\boldsymbol{a}-\boldsymbol{\mu}) = -\Sigma^{-1}(\boldsymbol{a}-\boldsymbol{\mu})(\boldsymbol{a}-\boldsymbol{\mu})^\top \Sigma^{-1}. \tag{12}$$

With these tools in hand, the following gradient expressions can be readily derived.

### A.1 GAUSSIAN

Being a member of the exponential family, the gradient of Gaussian log-likelihood allows straightforward derivation by using Eq. (10)-Eq. (12):

$$
\begin{aligned}
\ln \pi_s(a) &= -\frac{N}{2} \ln 2\pi - \frac{1}{2} \ln |\Sigma| - \frac{1}{2}(\boldsymbol{a} - \boldsymbol{\mu})^\top \Sigma^{-1}(\boldsymbol{a} - \boldsymbol{\mu}) \\
\Rightarrow \quad \nabla_{\boldsymbol{\mu}} \ln \pi_s(a) &= -\Sigma^{-1}(\boldsymbol{a} - \boldsymbol{\mu}), \\
\nabla_\Sigma \ln \pi_s(a) &= -\frac{1}{2}\left(\Sigma^{-1} - \Sigma^{-1}(\boldsymbol{a} - \boldsymbol{\mu})(\boldsymbol{a} - \boldsymbol{\mu})^\top \Sigma^{-1}\right).
\end{aligned}
\tag{13}
$$

### A.2 STUDENT'S T

In addition to $\boldsymbol{\mu}, \Sigma$, Student's t policy has an additional learnable parameter degree of freedom $\nu$. Recall that $\nu = 1$ corresponds to the Cauchy's distribution, while numerically with $\nu \geq 30$ it can be seen as a Gaussian distribution.

$$
\begin{aligned}
\ln \pi_s(a) &= \ln \Gamma\left(\frac{N+\nu}{2}\right) - \ln \Gamma\left(\frac{\nu}{2}\right) - \frac{N}{2}\ln \nu\pi - \frac{1}{2}\ln|\Sigma| - \frac{N+\nu}{2}\ln\left(1 + \frac{1}{\nu}(\boldsymbol{a}-\boldsymbol{\mu})^\top\Sigma^{-1}(\boldsymbol{a}-\boldsymbol{\mu})\right) \\
\Rightarrow \quad \nabla_{\boldsymbol{\mu}} \ln \pi_s(a) &= \frac{N+\nu}{\nu} \cdot \frac{\Sigma^{-1}(\boldsymbol{a}-\boldsymbol{\mu})}{1 + \frac{1}{\nu}(\boldsymbol{a}-\boldsymbol{\mu})^\top\Sigma^{-1}(\boldsymbol{a}-\boldsymbol{\mu})}, \\
\nabla_\Sigma \ln \pi_s(a) &= -\frac{1}{2}\left(\Sigma^{-1} - \frac{(N+\nu)\Sigma^{-1}(\boldsymbol{a}-\boldsymbol{\mu})(\boldsymbol{a}-\boldsymbol{\mu})^\top\Sigma^{-1}}{\nu + (\boldsymbol{a}-\boldsymbol{\mu})^\top\Sigma^{-1}(\boldsymbol{a}-\boldsymbol{\mu})}\right), \\
\nabla_\nu \ln \pi_s(a) &= \psi\left(\frac{N+\nu}{2}\right) - \psi\left(\frac{\nu}{2}\right) - \frac{N}{2\nu} - \frac{N}{2}\ln\left(1 + \frac{1}{\nu}(\boldsymbol{a}-\boldsymbol{\mu})^\top\Sigma^{-1}(\boldsymbol{a}-\boldsymbol{\mu})\right) \\
&\quad + \frac{N+\nu}{2}\frac{\frac{1}{\nu}(\boldsymbol{a}-\boldsymbol{\mu})^\top\Sigma^{-1}(\boldsymbol{a}-\boldsymbol{\mu})}{\nu + (\boldsymbol{a}-\boldsymbol{\mu})^\top\Sigma^{-1}(\boldsymbol{a}-\boldsymbol{\mu})},
\end{aligned}
\tag{14}
$$

where $\psi(\cdot)$ is the digamma function. For $\boldsymbol{\mu}$ and $\Sigma$ we again leveraged Eq. (10)-Eq. (12).

### A.3 $q$-GAUSSIAN

Since we do not parametrize the entropic index $q$, the gradients of log-likelihood with respect to $\boldsymbol{\mu}, \Sigma$ are the same for both heavy- and light-tailed $q$-Gaussian. Therefore, we focus on the light-tailed case $q < 1$ and absorb into the constant $C$ the terms only related to $q$.

$$
\begin{aligned}
\ln \pi_s(a) &= \ln C - \frac{1}{2}\ln|\Sigma| + \frac{1}{1-q}\ln\left[1 - \frac{1-q}{2}(\boldsymbol{a}-\boldsymbol{\mu})^\top\Sigma^{-1}(\boldsymbol{a}-\boldsymbol{\mu})\right]_+ \\
\Rightarrow \nabla_{\boldsymbol{\mu}} \ln \pi_s(a) &= \frac{1}{1-q}\frac{(1-q)\Sigma^{-1}(\boldsymbol{a}-\boldsymbol{\mu})}{\left[1 - \frac{1-q}{2}(\boldsymbol{a}-\boldsymbol{\mu})^\top\Sigma^{-1}(\boldsymbol{a}-\boldsymbol{\mu})\right]_+} = \frac{\Sigma^{-1}(\boldsymbol{a}-\boldsymbol{\mu})}{\exp_q\left(-\frac{1}{2}(\boldsymbol{a}-\boldsymbol{\mu})^\top\Sigma^{-1}(\boldsymbol{a}-\boldsymbol{\mu})\right)^{1-q}}, \\
\nabla_\Sigma \ln \pi_s(a) &= -\frac{1}{2}\left(\Sigma^{-1} - \frac{\Sigma^{-1}(\boldsymbol{a}-\boldsymbol{\mu})(\boldsymbol{a}-\boldsymbol{\mu})^\top\Sigma^{-1}}{\exp_q\left(-\frac{1}{2}(\boldsymbol{a}-\boldsymbol{\mu})^\top\Sigma^{-1}(\boldsymbol{a}-\boldsymbol{\mu})\right)^{1-q}}\right).
\end{aligned}
\tag{15}
$$

It is interesting to see that the gradients of $q$-Gaussian log-likelihood can be seen as the Gaussian counterparts scaled by the reciprocal of $\exp_q(\cdot)^{1-q}$. Since $\exp_q$ can take on zero values when $q < 1$, the gradients as well as the log-likelihood function may be undefined outside the support. However, this does not happen for heavy-tailed $q$-Gaussian $1 < q < 3$.

To make these policies suitable for deep reinforcement learning, we discuss in Appendix D how to parametrize the policies using neural networks.

## B CONNECTION TO ENTROPY REGULARIZATION

The $q$-exp family provides a general class of stochastic policies. But perhaps more importantly, they can be derived as solutions to the maximum Tsallis entropy principle (Suyari & Tsukada, 2005;

Furuichi, 2010), generalizing the maximum Shannon entropy principle (Jaynes, 1957; Grünwald & Dawid, 2004; Ziebart, 2010). We discuss both principles in equation 16.

For notational convenience, we define the inner product for any two functions $F_1, F_2 \in \mathbb{R}^{|\mathcal{S}| \times |\mathcal{A}|}$ over actions as $\langle F_1, F_2 \rangle \in \mathbb{R}^{|\mathcal{S}|}$. We write $F_s$ to express the function's dependency $F$ on state $s$. Often $F_s \in \mathbb{R}^{|\mathcal{A}|}$, whenever its component is of concern, we denote it by $F_s(a)$.

### B.1 BOLTZMANN-GIBBS REGULARIZATION

Consider a regularized policy as the solution to the following regularization problem:
$$\pi_{\Omega,s} = \arg\max_{\pi_s \in \Delta_{\mathcal{A}}} \langle \pi_s, Q_s \rangle - \Omega(\pi_s), \tag{16}$$
where $\Omega$ is a proper, lower semi-continuous, strictly convex function. We can absorb the regularization coefficient $\tau > 0$ into $\Omega$ by $\Omega := \tau \tilde{\Omega}$. It is a classic result that at the limit $\tau \to 0$ the unregularized optimal action is recovered: $\lim_{\tau \to 0} \pi_{\tau \tilde{\Omega}, s} = \mathbb{1}\{a = a^*\}$, i.e., $a^*$ that maximizes $Q_s$.

One of the most well-studied regularizers is the negative Shannon entropy $\Omega(\pi_s) = \langle \pi_s, \ln \pi_s \rangle$, which leads to the Boltzmann-Gibbs policy $\pi_{BG,s}(a) = \exp\left(Q_s(a) - Z_s\right)$. Another popular choice is the KL divergence $\Omega(\pi_s) = \langle \pi_s, \ln \pi_s - \ln \mu_s \rangle$ for some reference policy $\mu_s$. The regularized policy is $\pi_{KL,s}(a) = \mu_s(a) \exp\left(Q_s(a) - Z_s\right)$. Notice that it is also a member of the exponential family by writing $\pi_{KL,s}(a) = \exp\left(Q_s(a) - Z_s + \ln \mu_s(a)\right)$.

### B.2 TSALLIS REGULARIZATION

Originally, the deformed logarithm function was introduced in the statistical physics to generalize the Shannon entropy by deforming the logarithm contained in it (Naudts, 2010). Consider replacing Shannon entropy in equation 16 with the negative Tsallis entropy $\Omega_q(\pi_s) = \frac{1}{q-1}\left(\langle \mathbf{1}, \pi_s^q \rangle - 1\right)$. It has been shown that $\Omega_q(\pi_s)$ leads to following regularized policy:
$$\pi_{\Omega_q,s}(a) = \exp_{2-q}\left(Q_s(a) - Z'_{q,s}\right). \tag{17}$$
We see that when $q = 2$, it recovers the sparsemax policy introduced in Section 3.2. As indicated by (Zhu et al., 2023), the effect of different $q \in (-\infty, 1)$ lies in the extent of thresholding. One can also consider regularization by the Tsallis KL divergence $D_{KL}^q(\pi_s \| \mu_s) := \left\langle \pi_s, -\ln_q \frac{\mu_s}{\pi_s} \right\rangle$ (Furuichi et al., 2004). Likewise to the KL case, $\mu$ is typically taken to be the last policy, in which the regularized policy is the product of two $q$-exp functions.

It is worth noting that there are other regularization functionals that can induce $q$-exp policies. One of the prominent examples is the $\alpha$-entropy/divergence, which can be defined by simply letting $p = \frac{1}{q}$ in $\Omega_q(\pi_s)$ (Peters et al., 2019; Belousov & Peters, 2019). It is shown in (Xu et al., 2022; 2023) that when $\alpha = -1$ it induces the sparsemax policy. Therefore, $q$-exp policies can also be viewed as solutions to the $\alpha$ regularization.

### B.3 TSALLIS ADVANTAGE WEIGHTED ACTOR CRITIC

An advantage of $q$-exp (resp. exp) policies is it may improve the consistency of algorithms that explicitly mimics a $q$-exp (resp. exp) policy. For example, Tsallis Advantage Weighted Actor Critic (TAWAC) proposed to use a light-tailed $q$-exp policy for offline learning (Zhu et al., 2024). However, TAWAC was implemented with Gaussian, which amounts to approximating a light-tailed distribution using one with infinite support. Let $\pi_{\mathcal{D}}$ denote the empirical behavior policy and $\mathcal{D}$ the offline dataset. TAWAC minimizes the following actor loss, where we ignore the parametrization of value functions:
$$\mathcal{L}(\phi) := \mathbb{E}_{s \sim \mathcal{D}}\left[D_{KL}(\pi_{TKL,s} \| \pi_{\phi,s})\right] = \mathbb{E}_{\substack{s \sim \mathcal{D} \\ a \sim \pi_{\mathcal{D}}}}\left[-\exp_{q'}\left(\frac{Q_s(a) - V_s}{\tau}\right) \ln \pi_{\phi,s}(a)\right], \tag{18}$$
where $\pi_{TKL,s}(a) \propto \pi_{\mathcal{D},s}(a) \exp_{q'}\left(\tau^{-1}\left(Q_s(a) - V_s\right)\right)$ denotes the Tsallis KL regularized policy. We can generalize TAWAC to online learning by simply changing the expectation to be w.r.t. arbitrary behavior policy. It is clear that depending on $q'$, choosing Gaussian as $\pi_{\phi}$ may incur inconsistency with the theory. A $q$-exp policy would be more suitable and could improve the performance. As evidenced by our experimental results, heavy tailed policies indeed further improve the performance of TAWAC by a large margin.

# C    ACTOR LOSSES

To help understand when exp-family policies (resp. $q$-exp) may be more preferable, we compare the actor loss functions of the algorithms in the experiment section.

## C.1    ONLINE ALGORITHMS

**Soft Actor-Critic.** SAC minimizes the following KL loss for the actor

$$\mathcal{L}_{\text{SAC}}(\phi) := \mathbb{E}_{s \sim \mathcal{B}} \left[ D_{KL}(\pi_\phi(\cdot|s) \, \| \, \pi_{\text{BG}}(\cdot|s)) \right] = \mathbb{E}_{s \sim \mathcal{B}} \left[ D_{KL} \left( \pi_\phi(\cdot|s) \, \middle\| \, \frac{\exp\left(\tau^{-1} Q(s, \cdot)\right)}{Z_s} \right) \right],$$

where states are sampled from replay buffer $\mathcal{B}$. The parametrized policy $\pi_\phi$ is projected to be close to the BG policy, therefore it is reasonable to expect that choosing $\pi_\phi$ from the exp-family may be more preferable. Depending on action values, BG can be skewed, multi-modal. Therefore, the symmetric, unimodal Gaussian may not be able to fully capture these characteristics.

**Greedy Actor-Critic.** GreedyAC maintains an additional proposal policy besides the actor. The proposal policy is responsible for producing actions from which the top $k\%$ of actions are used to update the actor. The proposal policy itself is updated similarly but with an entropy bonus encouraging exploration. To simplify notations, we use $I(s)$ to denote the set containing top $k\%$ actions given $s$.

$$\mathcal{L}_{\text{GreedyAC, prop}}(\phi) := \mathbb{E}_{\substack{s \sim \mathcal{B} \\ a \in I(s)}} \left[ -\ln \pi_\phi(a|s) - \mathcal{H}\left(\pi_\phi(\cdot|s)\right) \right],$$

$$\mathcal{L}_{\text{GreedyAC, actor}}(\bar{\phi}) := \mathbb{E}_{\substack{s \sim \mathcal{B} \\ a \in I(s)}} \left[ -\ln \pi_{\bar{\phi}}(a|s) \right].$$

GreedyAC maximizes log-likelihood of the actor and proposal policy. These policies impose no constraints on the functional form of $\pi$.

**Online Tsallis AWAC.** Online TAWAC is extended to condition on the behavior policy that collects experiences $\pi_{\text{theory}}(a|s) \propto \pi_{\text{behavior}}(a|s) \exp_q \left( \frac{Q(s,a) - V(s)}{\tau} \right)$.

$$\mathcal{L}_{\text{TAWAC}}(\phi) := \mathbb{E}_{s \sim \mathcal{B}} \left[ D_{KL}(\pi_{\text{theory}}(\cdot|s) \, \| \, \pi_\phi(\cdot|s)) \right]$$

$$= \mathbb{E}_{\substack{s \sim \mathcal{B} \\ a \sim \pi_{\bar{\phi}}}} \left[ -\exp_q \left( \frac{Q(s, a) - V(s)}{\tau} \right) \ln \pi_\phi(a|s) \right],$$

where the condition $a \sim \pi_{\bar{\phi}}$ is because the target policy is used to sample actions. Since Tsallis AWAC explicitly minimizes KL loss to a $q$-exp policy, which can be light-tailed/heavy-tailed depending on $q$. Therefore, choosing a $q$-exp $\pi_\phi$ could lead to better performance.

## C.2    OFFLINE ALGORITHMS

**AWAC.** Advantage Weighted Actor-Critic (AWAC) is the basis of many algorithms. AWAC minimizes the following actor loss:

$$\mathcal{L}_{\text{AWAC}}(\phi) := \mathbb{E}_{\substack{s \sim \mathcal{D} \\ a \sim \pi_{\mathcal{D}}}} \left[ -\exp \left( \frac{Q(s, a) - V(s)}{\tau} \right) \ln \pi_\phi(a|s) \right],$$

which is derived as the result of minimizing KL loss $D_{KL}(\pi_{\mathcal{D}} \, \| \, \pi_\phi)$ and applying the trick in Eq. 18, i.e., $\pi_{\text{theory}}(a|s) \propto \pi_{\mathcal{D}}(a|s) \exp \left( \frac{Q(s,a) - V(s)}{\tau} \right) = \exp \left( \frac{Q(s,a) - V(s)}{\tau} - \ln \pi_{\mathcal{D}}(a|s) \right)$. However, the shape of this policy can be multi-modal and skewed depending on the values and $\pi_{\mathcal{D}}$. It is visible from experimental results that Beta and Squashed Gaussian have similar performance.

**IQL.** In contrast to AWAC, Implicit Q-Learning (IQL) does not have an explicit actor learning procedure and uses $\mathcal{L}_{\text{AWAC}}(\phi)$ as a means for policy extraction from the learned value functions. The exponential advantage function acts simply as weights. Therefore, IQL does not assume the functional form of $\pi_\phi$.

**InAC.** In-Sample Actor-Critic (InAC) proposed to impose an in-sample constraint on the entropy-regularized BG policy. As such, the dependence on the behavior policy is moved into the exponential-advantage weighting function:

$$\mathcal{L}_{\text{InAC}}(\phi) := \mathbb{E}_{\substack{s \sim \mathcal{D} \\ a \sim \pi_{\mathcal{D}}}} \left[ -\exp \left( \frac{Q(s, a) - V(s)}{\tau} - \ln \pi_{\mathcal{D}}(a|s) \right) \ln \pi_\phi(a|s) \right].$$

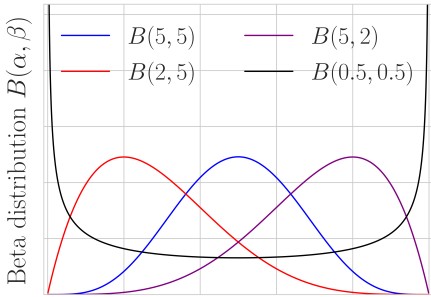

Figure 10: Beta distribution with $\alpha < 1, \beta < 1$ takes on a bowl shape rather than a bell shape. The shape can also be skewed as well as symmetric.

As a result, InAC is not as sensitive to the advantage weighting as AWAC does, which implies that InAC may favor an $\exp \pi_\phi$ but less than AWAC.

**Offline Tsallis AWAC.** The offline case of Tsallis AWAC is same as the online case except the change of expectation:

$$\mathcal{L}_{\text{TAWAC}}(\phi) := \mathbb{E}_{\substack{s \sim \mathcal{D} \\ a \sim \pi_{\mathcal{D}}}} \left[ -\exp_q \left( \frac{Q(s,a) - V(s)}{\tau} \right) \ln \pi_\phi(a|s) \right].$$

Same with the online case, offline Tsallis AWAC may theoretically prefer a $q$-exp $\pi_\phi$.

**TD3BC.** In Appendix E we include additional results of TD3BC (Fujimoto & Gu, 2021), whose actor loss is obtained by simply augmenting the TD3 loss with a behavior cloning term:

$$\mathcal{L}_{\text{TD3BC}}(\phi) := \mathbb{E}_{\substack{s \sim \mathcal{D} \\ a \sim \pi_{\mathcal{D}}}} \left[ \lambda Q(s, \pi(s)) - (\pi(s) - a)^2 \right].$$

The behavior cloning term is simply minimizing the $L_2$ distance to actions in the dataset. Though another interpretation by (Xiao et al., 2023) is that this term can be understood as applying KL regularization to Gaussian policy.

## D  IMPLEMENTATION DETAILS

Details of our implementation is provided in this section. Specifically, we detail our design choices hyperparameters and network architectures.

### D.1  POLICIES

We discuss how to parametrize Beta, Student's t and $q$-Gaussian policies. Specifically, we parametrize $\alpha, \beta$ for Beta policy; $\boldsymbol{\mu}, \Sigma$ for $q$-Gaussian. In additional to location and scale, Student's t has an additional learnable parameter $\nu$.

For Student's t policy, we initialized a base DOF $\nu_0 = 1$ and learn $\nu$ by the softplus function. The Student's t policy therefore always has DOF $\nu > 1$, which is equivalent to starting as the Cauchy's distribution. For Beta policy, we similarly constrain $\alpha, \beta$ to be the output of softplus function plus 1. This is because when $\alpha < 1, \beta < 1$ the Beta policy takes on a bowl shape rather than a bell shape, see Figure 10. For Gaussian and $q$-Gaussian policies, we follow the standard practice to parametrize mean by the tanh activation and scale by the log-std transform.

In the tested off-policy/offline algorithms, it is necessary to evaluate log-probability for off-policy/offline actions stored in the buffer. For light-tailed $q$-Gaussian this can cause numerical issues since the evaluated actions may fall outside the support, incurring $-\infty$ for log-probability. To avoid this issue, we sample a batch of on-policy actions from the $q$-Gaussian and replace the out-of-support actions with the nearest action in the $L_2$ sense.

In our experiments, all environments had bounded action space. Squashed Gaussian and light-tailed q-Gaussian provide bounded output. However, Student's t, heavy tailed q-Gaussian and Gaussian have unbounded support. For these distributions, we clipped the sampled action to fit the action space of the task, without further modification on the density. The mean value is constrained using tanh function in distributions with unbounded support, except the standard Gaussian in offline learning.

## D.2 Online Experiments

We used three classical control environments in the continuous action setting: Mountain Car (Sutton & Barto, 2018), Pendulum (Degris et al., 2012) and Acrobot (Sutton & Barto, 2018). All episodes are truncated at 1000 time steps. In Mountain Car, the action is the force applied to the car in $[-1, 1]$, and the agent receives a reward of -1 at every time step. In Pendulum, the action is the torque applied to the base of the pendulum in $[-2, 2]$ and the reward is defined by $r = -(\theta^2 + 0.1 * (\frac{d\theta}{dt})^2 + 0.001 * a^2)$ where $\theta$ denotes the angle, $\frac{d\theta}{dt}$ is the derivative of time and $a$ the torque applied. Finally, in acrobot, the action is the torque applied on the joint between two links in $[-1, 1]$ and the agent receives a reward of $-1$ per time step.

**Experiment settings:** When sweeping different hyperparameter configurations, we pause the training every 10,000 time steps and then evaluate the learned policy by averaging the total reward over 3 episodes. However, when running the best hyperparameter configuration, we evaluate by freezing the policy every 1000 time steps and then computing the total reward obtained for 1 episode.

**Parameter sweeping:** We sweep the hyperparameters with 5 independent runs and then evaluate the run configuration for 30 seeds. We select the best hyperparameters based on the overall area under curve. When running the best hyperparameter configurations, we discard the original 5 seeds used for the hyperparameter sweep in order to avoid the bias caused by hyperparameter selection. Details regarding the fixed and swept hyperparameters are provided in Table 4.

**Agent learning:** We used a 2-layer network with 64 nodes on each layer and ReLU non-linearities. The batch size was 32. Agents used a target network for the critic, updated with polyak averaging with $\alpha = 0.01$.

| Hyperparameter | Value |
|---|---|
| Critic Learning rate | Swept in $\{1 \times 10^{-2}, 1 \times 10^{-3}, 1 \times 10^{-4}, 1 \times 10^{-5}\}$ |
| Critic learning rate multiplier for actor | Swept in $\{0.1, 1, 10\}$ |
| Temperature | Swept in $\{0.01, 0.1, 1\}$ |
| Discount rate | 0.99 |
| Hidden size of Value network | 64 |
| Hidden layers of Value network | 2 |
| Hidden size of Policy network | 64 |
| Hidden layers of Policy network | 2 |
| Minibatch size | 32 |
| Adam.$\beta_1$ | 0.9 |
| Adam.$\beta_2$ | 0.999 |
| Number of seeds for sweeping | 10 |
| Number of seeds for the best setting | 30 |

Table 4: Default hyperparameters and sweeping choices for online experiments.

## D.3 Offline Experiments

We use the MuJoCo suite from D4RL (Apache-2/CC-BY licence) (Fu et al., 2020) for offline experiments. The D4RL offline datasets all contain 1 million samples generated by a partially trained SAC agent. The name reflects the level of the trained agent used to collect the transitions. The Medium dataset contains samples generated by a medium-level (trained halfway) SAC policy. Medium-expert mixes the trajectories from the Medium level and that produced by an expert agent. Medium-replay consists of samples in the replay buffer during training until the policy reaches the

medium level of performance. In summary, the ranking of levels is Medium-expert > Medium > Medium-replay.

**Experiment settings:** We conducted the offline experiment using 9 datasets provided in D4RL: halfcheetah-medium-expert, halfcheetah-medium, halfcheetah-medium-replay, hopper-medium-expert, hopper-medium, hopper-medium-replay, walker2d-medium-expert, walker2d-medium, and walker2d-medium-replay. We run 5 agents: TAWAC, AWAC, IQL, InAC, and TD3BC. The results of TD3BC are posted in the appendix. For each agent, we tested 5 distributions: Gaussian, Squashed Gaussian, Beta, Student's t, and Heavy-tailed $q$-Gaussian. As offline learning algorithms usually require a distribution covering the whole action space, Light-tailed q-Gaussian is not considered in offline learning experiments. Each agent was trained for $1 \times 10^6$ steps. The policy was evaluated every 1000 steps. The score was averaged over 5 rollouts in the real environment; each had 1000 steps.

**Parameter sweeping:** All results shown in the paper were generated by the best parameter setting after sweeping. We list the parameter setting in Table 5. Learning rate and temperature in TAWAC + medium datasets were swept as the experiments in their publication did not include the medium dataset. The best learning rates are reported in Table 6, and the temperatures are listed in Table 7.

| Hyperparameter | Value |
|---|---|
| Learning rate | Swept in $\{3 \times 10^{-3}, 1 \times 10^{-3}, 3 \times 10^{-4}, 1 \times 10^{-4}\}$ See the best setting in Table 6 |
| Temperature | Same as the number reported in the publication of each algorithm. Except in TAWAC + medium datasets, the value was swept in $\{1.0, 0.5, 0.01\}$. See the setting in Table 7 |
| IQL Expectile | 0.7 |
| Discount rate | 0.99 |
| Hidden size of Value network | 256 |
| Hidden layers of Value network | 2 |
| Hidden size of Policy network | 256 |
| Hidden layers of Policy network | 2 |
| Minibatch size | 256 |
| Adam.$\beta_1$ | 0.9 |
| Adam.$\beta_2$ | 0.99 |
| Number of seeds for sweeping | 5 |
| Number of seeds for the best setting | 10 |

Table 5: Default hyperparameters and sweeping choices for offline experiments.

**Agent learning:** We used a 2-layer network with 256 nodes on each layer. The batch size was 256. Agents used a target network for the critic, updated with polyak averaging with $\alpha = 0.005$. The discount rate was set to 0.99.

**Sampling.** To give an intuition for sampling time, we drew $10^5$ samples from a randomly initialized actor on two environments: HalfCheetah with 17-dim state and 6-dim action. The sparse $q$-Gaussian, heavy-tailed $q$-Gaussian and Gaussian respectively cost (107.12, 72.09, 27.94) seconds. We confirmed that the methods in Alg. 1 were on the same magnitude to the Gaussian, but the sparse $q$-Gaussian cost more than the heavy-tailed due to more computation to produce low-variance samples. This is further confirmed by Hopper with 11-dim state, 3-dim action, where they costed (98.13, 65.17, 25.17) seconds.

## E   FURTHER RESULTS

Figure 11 shows the Manhattan plot of Soft-Actor-Critic (SAC) with all swept hyperparameters on the online classic control environments. Student-t and Gaussian both seem to have a similar behavior to hyperparameters. Although there is no definitive winner here, we can safely conclude that if we have a problem where Gaussian works, Student-t is very likely to work. Additionally, give the results

| Dataset | Distribution | TAWAC | AWAC | IQL | InAC | TD3BC |
|---|---|---|---|---|---|---|
| HalfCheetah-Medium-Expert | Heavy-Tailed q-Gaussian | 0.001 | 0.001 | 0.001 | 0.001 | 0.0003 |
| HalfCheetah-Medium-Expert | Squashed Gaussian | 0.001 | 0.0003 | 0.0003 | 0.001 | 0.0003 |
| HalfCheetah-Medium-Expert | Gaussian | 0.0003 | 0.0001 | 0.0003 | 0.0003 | 0.0003 |
| HalfCheetah-Medium-Expert | Beta | 0.001 | 0.0003 | 0.001 | 0.001 | 0.001 |
| HalfCheetah-Medium-Expert | Student's t | 0.001 | 0.0003 | 0.0003 | 0.0003 | 0.001 |
| HalfCheetah-Medium-Replay | Heavy-Tailed q-Gaussian | 0.001 | 0.001 | 0.001 | 0.001 | 0.001 |
| HalfCheetah-Medium-Replay | Squashed Gaussian | 0.001 | 0.0003 | 0.0003 | 0.001 | 0.003 |
| HalfCheetah-Medium-Replay | Gaussian | 0.001 | 0.0001 | 0.0003 | 0.001 | 0.001 |
| HalfCheetah-Medium-Replay | Beta | 0.001 | 0.0003 | 0.0003 | 0.001 | 0.001 |
| HalfCheetah-Medium-Replay | Student's t | 0.001 | 0.0003 | 0.0003 | 0.0003 | 0.003 |
| HalfCheetah-Medium | Heavy-Tailed q-Gaussian | 0.001 | 0.001 | 0.001 | 0.001 | 0.0003 |
| HalfCheetah-Medium | Squashed Gaussian | 0.001 | 0.0003 | 0.001 | 0.001 | 0.0003 |
| HalfCheetah-Medium | Gaussian | 0.0003 | 0.0001 | 0.0003 | 0.001 | 0.001 |
| HalfCheetah-Medium | Beta | 0.001 | 0.001 | 0.001 | 0.001 | 0.0003 |
| HalfCheetah-Medium | Student's t | 0.001 | 0.0003 | 0.001 | 0.001 | 0.001 |
| Hopper-Medium-Expert | Heavy-Tailed q-Gaussian | 0.001 | 0.001 | 0.001 | 0.001 | 0.0001 |
| Hopper-Medium-Expert | Squashed Gaussian | 0.001 | 0.001 | 0.001 | 0.001 | 0.0001 |
| Hopper-Medium-Expert | Gaussian | 0.0003 | 0.0003 | 0.001 | 0.001 | 0.0001 |
| Hopper-Medium-Expert | Beta | 0.001 | 0.001 | 0.001 | 0.003 | 0.0003 |
| Hopper-Medium-Expert | Student's t | 0.001 | 0.0003 | 0.001 | 0.003 | 0.0001 |
| Hopper-Medium-Replay | Heavy-Tailed q-Gaussian | 0.001 | 0.0001 | 0.001 | 0.0001 | 0.001 |
| Hopper-Medium-Replay | Squashed Gaussian | 0.0001 | 0.0003 | 0.001 | 0.0003 | 0.001 |
| Hopper-Medium-Replay | Gaussian | 0.0003 | 0.0003 | 0.001 | 0.0003 | 0.001 |
| Hopper-Medium-Replay | Beta | 0.0001 | 0.0003 | 0.0003 | 0.003 | 0.003 |
| Hopper-Medium-Replay | Student's t | 0.0003 | 0.0003 | 0.0003 | 0.0003 | 0.001 |
| Hopper-Medium | Heavy-Tailed q-Gaussian | 0.003 | 0.001 | 0.001 | 0.001 | 0.0001 |
| Hopper-Medium | Squashed Gaussian | 0.001 | 0.0003 | 0.001 | 0.0003 | 0.0001 |
| Hopper-Medium | Gaussian | 0.001 | 0.001 | 0.0003 | 0.001 | 0.001 |
| Hopper-Medium | Beta | 0.001 | 0.001 | 0.003 | 0.001 | 0.001 |
| Hopper-Medium | Student's t | 0.001 | 0.001 | 0.001 | 0.001 | 0.0001 |
| Walker2d-Medium-Expert | Heavy-Tailed q-Gaussian | 0.0003 | 0.001 | 0.001 | 0.001 | 0.0003 |
| Walker2d-Medium-Expert | Squashed Gaussian | 0.001 | 0.001 | 0.0003 | 0.001 | 0.0003 |
| Walker2d-Medium-Expert | Gaussian | 0.0003 | 0.0001 | 0.0003 | 0.001 | 0.001 |
| Walker2d-Medium-Expert | Beta | 0.001 | 0.0003 | 0.001 | 0.001 | 0.001 |
| Walker2d-Medium-Expert | Student's t | 0.001 | 0.0003 | 0.0003 | 0.0003 | 0.0003 |
| Walker2d-Medium-Replay | Heavy-Tailed q-Gaussian | 0.0003 | 0.0003 | 0.003 | 0.0003 | 0.001 |
| Walker2d-Medium-Replay | Squashed Gaussian | 0.001 | 0.0003 | 0.0003 | 0.001 | 0.001 |
| Walker2d-Medium-Replay | Gaussian | 0.001 | 0.0003 | 0.0003 | 0.001 | 0.003 |
| Walker2d-Medium-Replay | Beta | 0.001 | 0.0003 | 0.0003 | 0.001 | 0.0003 |
| Walker2d-Medium-Replay | Student's t | 0.0003 | 0.0003 | 0.0003 | 0.001 | 0.001 |
| Walker2d-Medium | Heavy-Tailed q-Gaussian | 0.003 | 0.001 | 0.001 | 0.001 | 0.0001 |
| Walker2d-Medium | Squashed Gaussian | 0.001 | 0.001 | 0.001 | 0.001 | 0.0001 |
| Walker2d-Medium | Gaussian | 0.001 | 0.0001 | 0.001 | 0.001 | 0.0001 |
| Walker2d-Medium | Beta | 0.001 | 0.0003 | 0.003 | 0.001 | 0.0001 |
| Walker2d-Medium | Student's t | 0.001 | 0.0003 | 0.001 | 0.001 | 0.0001 |

Table 6: Best learning rates for offline experiments.

in the main text, Student's t more likely to perform better given the same hyperparameter sweeping range.

Our additional offline results include all algorithm-policy combination on all environments. We also include TD3BC (Fujimoto & Gu, 2021) for comparison. Figure 12 shows the overall comparison with TD3. It is clear that Squashed Gaussian performs well and Beta can show slight improvements in some cases. Though it is visible that no much difference is shown except on the Medium-Replay data. We conjecture that the better performance of Squashed Gaussian and Beta could be due to the TD3BC behavior cloning loss. It is encouraged that policy closely approximates the actions from the dataset. Therefore, policies like Beta that can concentrate faster may be more advantageous.

Figures 13 to 15 display boxplots of the combinations on environments of each level. Consistent observations to that in the main text can be drawn from these plots, but with the exception that in Figure 14 the environment-wise best combination is TAWAC + Student's t. TD3BC does not exhibit strong sensitivity to the choice of policy.

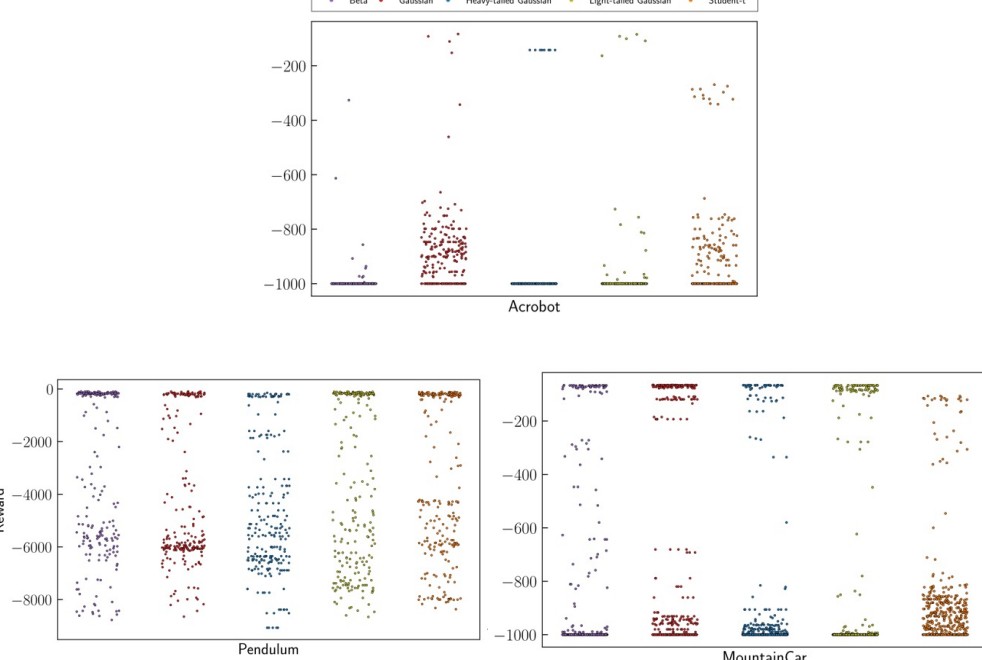

Figure 11: Manhattan plot of Soft-Actor-Critic (SAC) with all swept hyperparameters on the online classic control environments. The rewards on the y-axis are averaged over the final 10% of the total steps. Since different policy parameterizations have different numbers of runs in the sweep, we oversampled the smaller sweeps with replacement. From the plot of Acrobot, we observe that Student-t and Gaussian both respond similarly to changing hyper-parameters. Therefore, we hypothesize that if we have an environment where the Gaussian policy works, Student-t is also very likely to work. Additionally, from Figure 5 (left), we know that student-t is 75% more likely to outperform the Gaussian given the same hyperparameter sweeping range.

| Dataset | Distribution | TAWAC | AWAC | IQL | InAC | TD3BC |
|---|---|---|---|---|---|---|
| HalfCheetah-Medium-Expert | Heavy-Tailed q-Gaussian | 1.00 | 1.00 | 0.33 | 0.10 | 2.50 |
| HalfCheetah-Medium-Expert | Squashed Gaussian | 1.00 | 1.00 | 0.33 | 0.10 | 2.50 |
| HalfCheetah-Medium-Expert | Gaussian | 1.00 | 1.00 | 0.33 | 0.10 | 2.50 |
| HalfCheetah-Medium-Expert | Beta | 1.00 | 1.00 | 0.33 | 0.10 | 2.50 |
| HalfCheetah-Medium-Expert | Student's t | 1.00 | 1.00 | 0.33 | 0.10 | 2.50 |
| HalfCheetah-Medium-Replay | Heavy-Tailed q-Gaussian | 0.01 | 1.00 | 0.33 | 0.50 | 2.50 |
| HalfCheetah-Medium-Replay | Squashed Gaussian | 0.01 | 1.00 | 0.33 | 0.50 | 2.50 |
| HalfCheetah-Medium-Replay | Gaussian | 0.01 | 1.00 | 0.33 | 0.50 | 2.50 |
| HalfCheetah-Medium-Replay | Beta | 0.01 | 1.00 | 0.33 | 0.50 | 2.50 |
| HalfCheetah-Medium-Replay | Student's t | 0.01 | 1.00 | 0.33 | 0.50 | 2.50 |
| HalfCheetah-Medium | Heavy-Tailed q-Gaussian | 0.01 | 0.50 | 0.33 | 0.33 | 2.50 |
| HalfCheetah-Medium | Squashed Gaussian | 0.01 | 0.50 | 0.33 | 0.33 | 2.50 |
| HalfCheetah-Medium | Gaussian | 0.01 | 0.50 | 0.33 | 0.33 | 2.50 |
| HalfCheetah-Medium | Beta | 0.01 | 0.50 | 0.33 | 0.33 | 2.50 |
| HalfCheetah-Medium | Student's t | 0.01 | 0.50 | 0.33 | 0.33 | 2.50 |
| Hopper-Medium-Expert | Heavy-Tailed q-Gaussian | 0.50 | 1.00 | 0.33 | 0.01 | 2.50 |
| Hopper-Medium-Expert | Squashed Gaussian | 0.50 | 1.00 | 0.33 | 0.01 | 2.50 |
| Hopper-Medium-Expert | Gaussian | 0.50 | 1.00 | 0.33 | 0.01 | 2.50 |
| Hopper-Medium-Expert | Beta | 0.50 | 1.00 | 0.33 | 0.01 | 2.50 |
| Hopper-Medium-Expert | Student's t | 0.50 | 1.00 | 0.33 | 0.01 | 2.50 |
| Hopper-Medium-Replay | Heavy-Tailed q-Gaussian | 0.50 | 0.50 | 0.33 | 0.50 | 2.50 |
| Hopper-Medium-Replay | Squashed Gaussian | 0.50 | 0.50 | 0.33 | 0.50 | 2.50 |
| Hopper-Medium-Replay | Gaussian | 0.50 | 0.50 | 0.33 | 0.50 | 2.50 |
| Hopper-Medium-Replay | Beta | 0.50 | 0.50 | 0.33 | 0.50 | 2.50 |
| Hopper-Medium-Replay | Student's t | 0.50 | 0.50 | 0.33 | 0.50 | 2.50 |
| Hopper-Medium | Heavy-Tailed q-Gaussian | 0.50 | 0.50 | 0.33 | 0.10 | 2.50 |
| Hopper-Medium | Squashed Gaussian | 0.50 | 0.50 | 0.33 | 0.10 | 2.50 |
| Hopper-Medium | Gaussian | 0.50 | 0.50 | 0.33 | 0.10 | 2.50 |
| Hopper-Medium | Beta | 0.50 | 0.50 | 0.33 | 0.10 | 2.50 |
| Hopper-Medium | Student's t | 0.01 | 0.50 | 0.33 | 0.10 | 2.50 |
| Walker2d-Medium-Expert | Heavy-Tailed q-Gaussian | 0.01 | 0.10 | 0.33 | 0.10 | 2.50 |
| Walker2d-Medium-Expert | Squashed Gaussian | 0.01 | 0.10 | 0.33 | 0.10 | 2.50 |
| Walker2d-Medium-Expert | Gaussian | 0.01 | 0.10 | 0.33 | 0.10 | 2.50 |
| Walker2d-Medium-Expert | Beta | 0.01 | 0.10 | 0.33 | 0.10 | 2.50 |
| Walker2d-Medium-Expert | Student's t | 0.01 | 0.10 | 0.33 | 0.10 | 2.50 |
| Walker2d-Medium-Replay | Heavy-Tailed q-Gaussian | 0.50 | 0.10 | 0.33 | 0.50 | 2.50 |
| Walker2d-Medium-Replay | Squashed Gaussian | 0.50 | 0.10 | 0.33 | 0.50 | 2.50 |
| Walker2d-Medium-Replay | Gaussian | 0.50 | 0.10 | 0.33 | 0.50 | 2.50 |
| Walker2d-Medium-Replay | Beta | 0.50 | 0.10 | 0.33 | 0.50 | 2.50 |
| Walker2d-Medium-Replay | Student's t | 0.50 | 0.10 | 0.33 | 0.50 | 2.50 |
| Walker2d-Medium | Heavy-Tailed q-Gaussian | 0.01 | 0.10 | 0.33 | 0.33 | 2.50 |
| Walker2d-Medium | Squashed Gaussian | 1.00 | 0.10 | 0.33 | 0.33 | 2.50 |
| Walker2d-Medium | Gaussian | 1.00 | 0.10 | 0.33 | 0.33 | 2.50 |
| Walker2d-Medium | Beta | 1.00 | 0.10 | 0.33 | 0.33 | 2.50 |
| Walker2d-Medium | Student's t | 0.01 | 0.10 | 0.33 | 0.33 | 2.50 |

Table 7: Temperature settings for offline experiments.

Table 8 examined the accumulated probabilities that fell on each It can be seen that the Student's t and the Gaussian tended to increasingly put more densities on the boundaries. This is in sheer contrast to the heavy-tailed $q$-Gaussian that put the majority of probability density within the boundary. This may explain the better performance of TAWAC + heavy-tailed $q$-Gaussian.

Lastly, for all of the results shown above, their learning curves are shown in Figures 16 to 20. We smoothed the curves with window size 10 for better visualization.

| Policy \ # Updates | 0 | 100 | 200 | 300 | 400 |
|---|---|---|---|---|---|
| Heavy-tailed $q$-Gaussian | (24.39, 13.19) | (45.23, 2.36) | (45.49, 2.04) | (45.52, 1.98) | (45.54, 1.89) |
| Student's t | (148.43, 71.23) | (198.89, 45.30) | (205.04, 37.04) | (207.15, 32.96) | (207.00, 33.84) |
| Gaussian | (190.96, 65.89) | (206.92, 53.05) | (211.77, 39.08) | (213.57, 33.71) | (214.39, 31.26) |

Table 8: The summation of probability density accumulated on the left and the right edge in Figure 9 before clipping. Each pair indicates the left and right edge. The Student's t and the Gaussian increasing put more densities on the edges as compared to the heavy-tailed $q$-Gaussian.

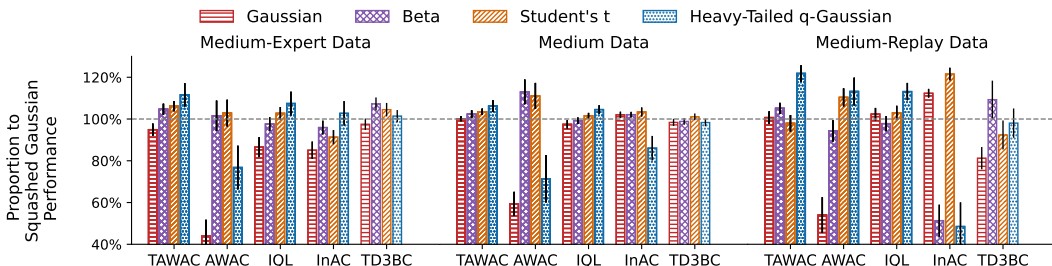

Figure 12: Relative improvement to the Squashed Gaussian policy, averaged over environments. Black vertical lines at the top indicate one standard error. For TD3BC, Beta policy outperforms the Squashed Gaussian on Medium-Expert and Medium-Replay.

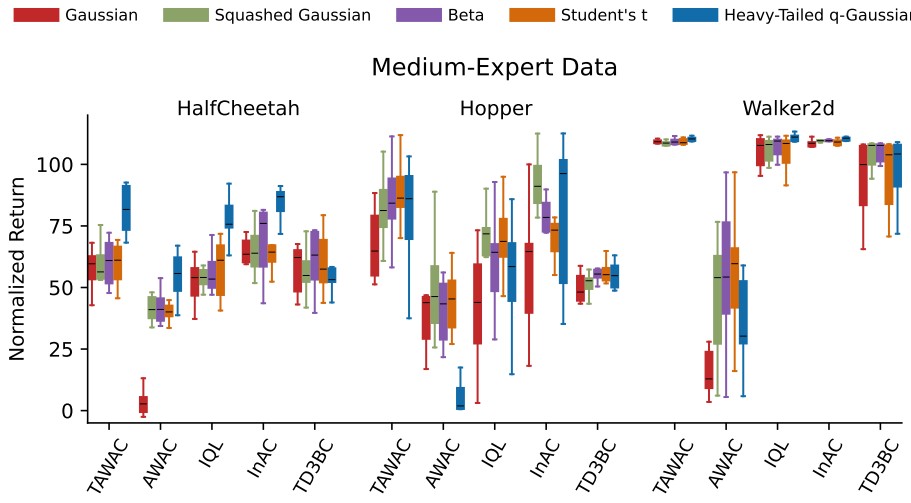

Figure 13: Normalized scores on Medium-Expert level datasets. The black bar shows the median. Boxes and whiskers show $1\times$ and $1.5\times$ interquartile ranges, respectively. Fliers are not plotted for uncluttered visualization. Environment-wise, InAC with heavy-tailed $q$-Gaussian is the top performer. Algorithm-wise, heavy-tailed or/and Student's t can improve or match the performance of the Squashed Gaussian except AWAC. With TD3BC no significant difference between policies is observed.

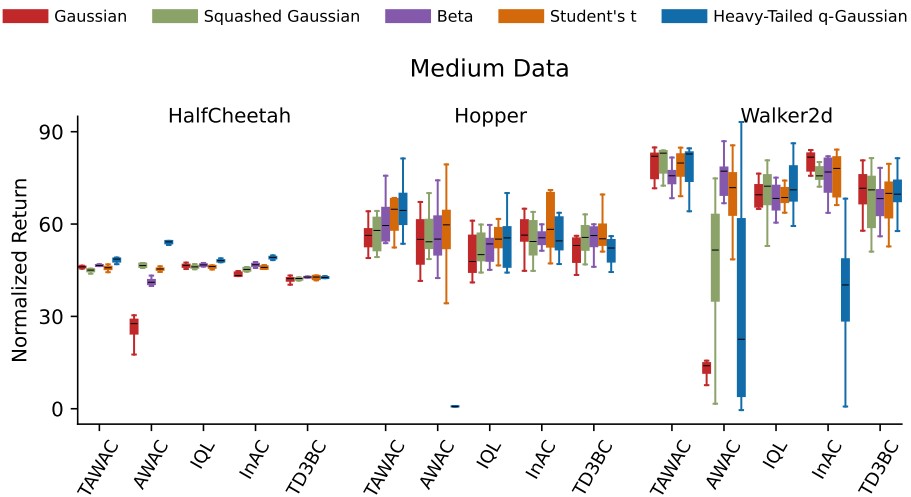

Figure 14: Normalized scores on Medium level datasets. The black bar shows the median. Boxes and whiskers show $1\times$ and $1.5\times$ interquartile ranges, respectively. Fliers are not plotted for uncluttered visualization. Environment-wise, InAC with heavy-tailed $q$-Gaussian is the top performer. Algorithm-wise, heavy-tailed $q$-Gaussian has observed significant performance drop with AWAC and InAC on Hopper and Walker2d. With TD3BC no significant difference between policies is observed.

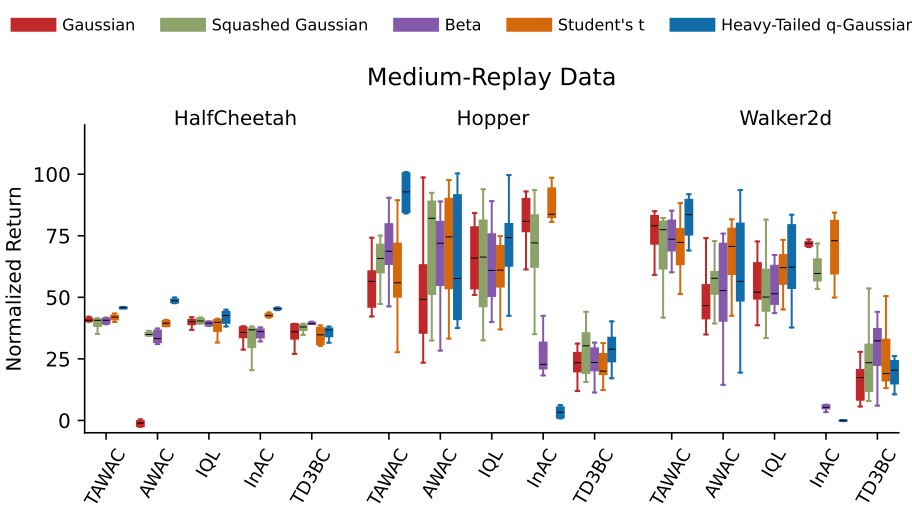

Figure 15: Normalized scores on Medium-Replay level datasets. The black bar shows the median. Boxes and whiskers show $1\times$ and $1.5\times$ interquartile ranges, respectively. Fliers are not plotted for uncluttered visualization. Environment-wise, TAWAC + heavy-tailed $q$-Gaussian is the best performer. Algorithm-wise, Student's t is stable and can match or improve on the performance of (Squashed) Gaussian.

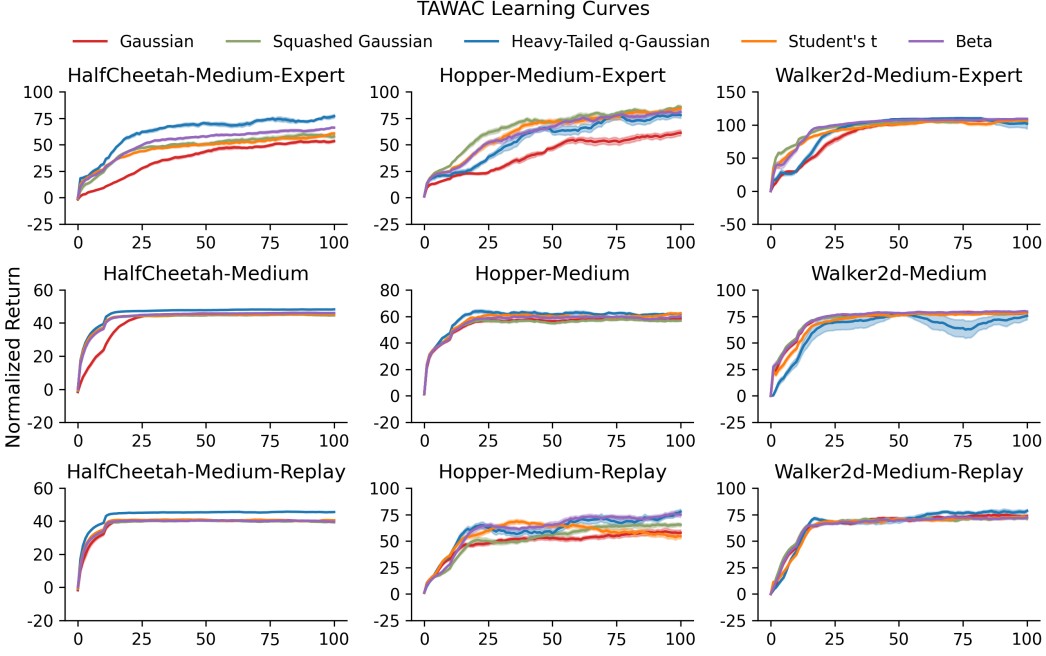

Figure 16: TAWAC learning curves in all datasets. Columns show different environments and rows are the levels of the environments. x-axis denotes the number of steps ($\times 10^4$), and y-axis is the normalized score. Each curve was smoothed with window size 10.

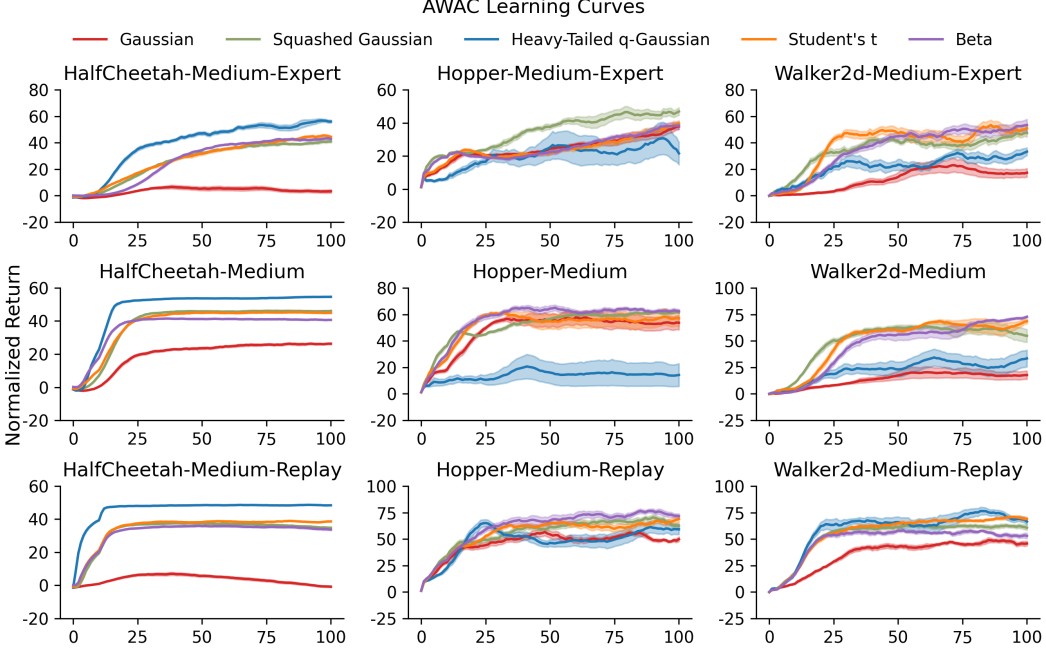

Figure 17: AWAC learning curves in all datasets. Columns show different environments and rows are the levels of the environments. x-axis denotes the number of steps ($\times 10^4$), and y-axis is the normalized score. Each curve was smoothed with window size 10.

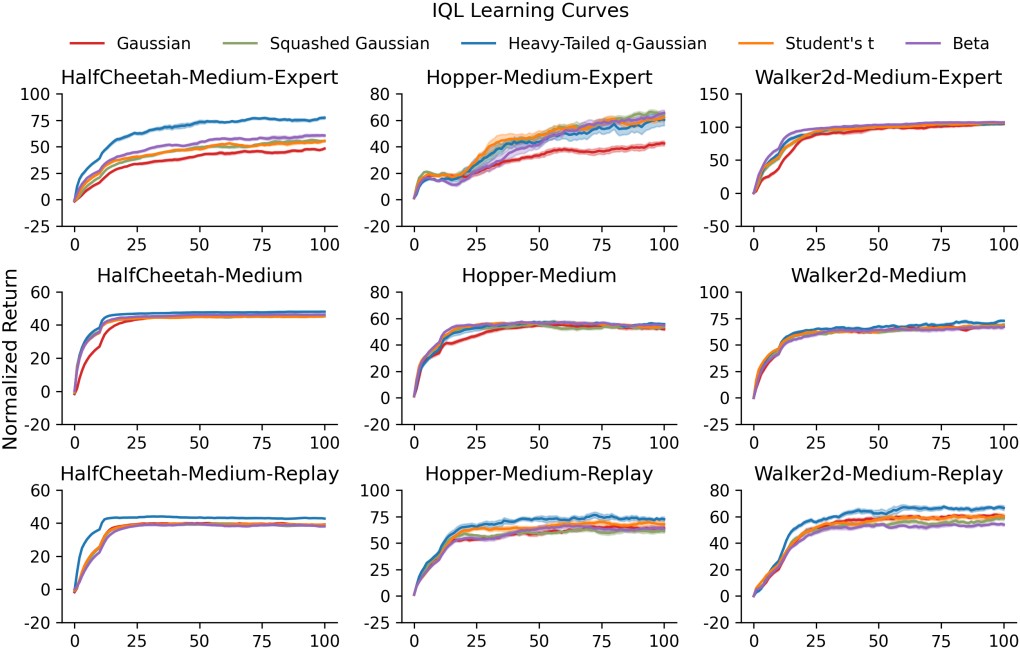

Figure 18: IQL learning curves in all datasets. Columns show different environments and rows are the levels of the environments. x-axis denotes the number of steps ($\times 10^4$), and y-axis is the normalized score. Each curve was smoothed with window size 10.

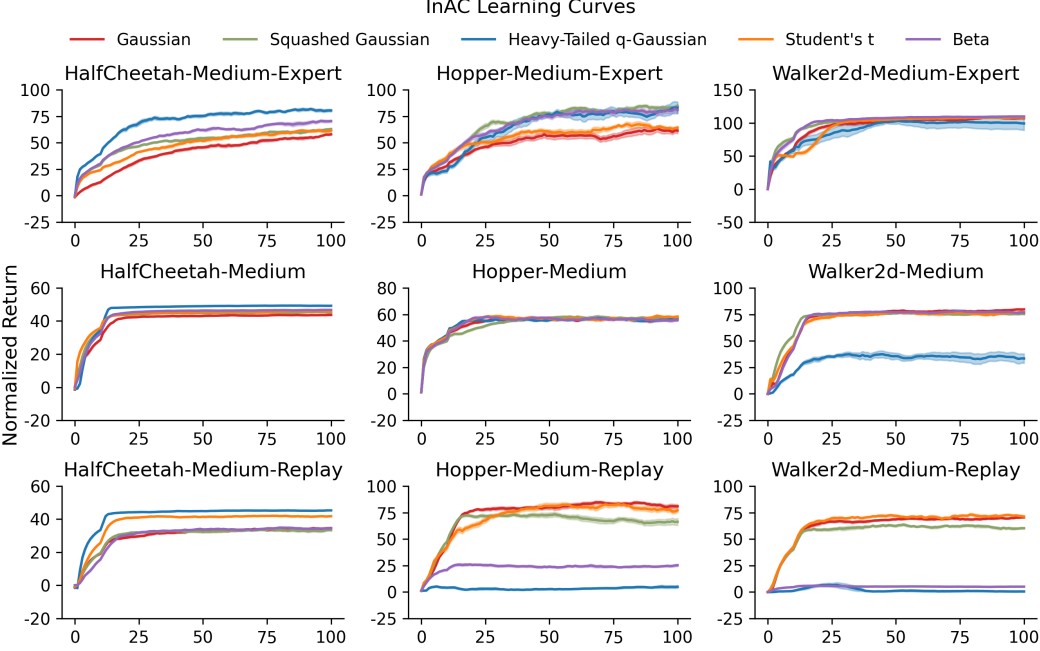

Figure 19: InAC learning curves in all datasets. Columns show different environments and rows are the levels of the environments. x-axis denotes the number of steps ($\times 10^4$), and y-axis is the normalized score. Each curve was smoothed with window size 10.

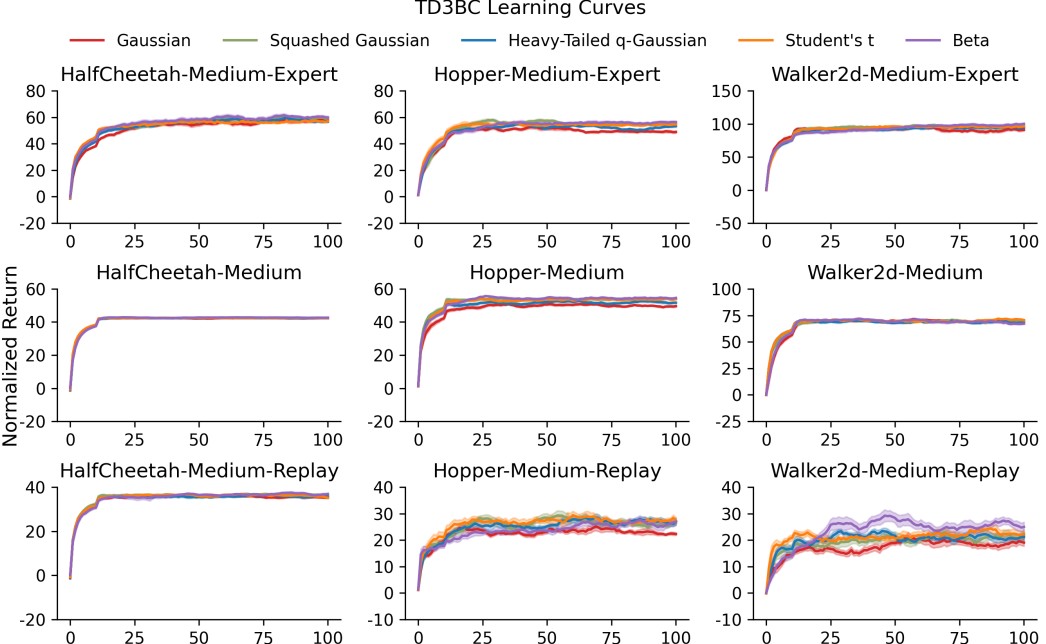

Figure 20: TD3+BC learning curves in all datasets. Columns show different environments and rows are the levels of the environments. x-axis denotes the number of steps ($\times 10^4$), and y-axis is the normalized score. Each curve was smoothed with window size 10.

