# OpenReview forum: "$q$-exponential family for policy optimization"
_ICLR.cc/2025/Conference — ICLR 2025 Poster_

### Official Review · Reviewer_Thwb · 2024-10-31

**Soundness:** 3
**Presentation:** 3
**Contribution:** 2
**Rating:** 6
**Confidence:** 2

**Summary:**

In continuous action RL many researches parameterize the policy as a mapping from states to std and mean of a Gaussian from which the actions are sampled.
When also the standard deviation of the Gaussian is learned, the authors argue that phenomenon like instabilities and lack of exploration can occurs. To this end, this paper explores empirically other continuous distribution to use to parameterize policies. In particular, it seems that heavier tails distributions are beneficial in Continuous Online Control experiments such as Acrobot and Cartpole and in Offline MuJoCo experiments.

**Strengths:**

I think that it is useful to have a paper that clearly states that the Gaussian parameterization might not be the best performing in practice.

**Weaknesses:**

The paper does not have a clear explanation or conclusive experiment about which parameterization should be used in which cases.
The main take away is that Gaussian parameterization should be avoided but it is not clear with which other distribution it should be replaced.

**Questions:**

Is it possible to understand which characteristics of the environment makes ine distribution better than the other ?


It would be very useful for the user to get some prior information about which distributions could work well to avoid to search over all the distributions proposed in your work and finding the best performing one.

---

> ### Author Response · Authors · 2024-11-20
> **added more results and explanations to conclude Student's t as the drop-in replacement to Gaussian**
>
> We thank the reviewer's insightful comments regarding conclusion of the paper.
>
> To help conclude that the Student's t as the policy to go, we have added the new Figure 11 showing the similar behavior of Student's t and Gaussian to hyperparameters. The following new paragraph in Section 5.1 states why Student's t can be used as a drop-in replacement to the Gaussian :
>
> > In Figure 11 we show the Manhattan plot of SAC with all swept hyperparameters on all environments. Though there is no a definitive winner for all cases, it is visible that the Student’s t and Gaussian have a similar behavior to hyperparameters. Therefore, if we are tackling a problem where the Gaussian works, the Student-t is very likely to work and perform better given the same hyperparameter sweeping range.
>
> We further consolidate our conclusion by adding the following remarks to the paper.
>
> > (Abstract) In summary, we find that the Student’s t policy a strong candidate for drop-in replacement to the Gaussian.
>
> > (Conclusion)  In summary, we found the Student’s t policy to be generally more performing and stable than the Gaussian and could be used as a drop-in replacement.
>
>
> There could be many factors of an environment that makes one distribution better than the others. For example, our Mountain Car has cost-to-go as reward that demands exploration. The ranking of policies seems very different to the environments with normal rewards. We believe decoupling the environmental characteristics for choosing a suitable policy parametrization is an interesting and important future direction.

---

> > ### Comment · Reviewer_Thwb · 2024-11-20
> >
> > Thank you for your answer !
> >
> > I keep my original score and continue supporting acceptance of this submission.
> >
> > Best,
> > Reviewer

---

### Official Review · Reviewer_dEFf · 2024-11-04

**Soundness:** 3
**Presentation:** 4
**Contribution:** 3
**Rating:** 8
**Confidence:** 4

**Summary:**

This paper conducts an empirical investigation of the use of the q-exponential family of policy parameterizations as an alternative to the commonly used Gaussian for continuous state action spaces in RL for policy optimization. In particular, these policies have been incorporated into existing Actor-Critic algorithms and experiments have been performed to compare the different policies to showcase the improved performance of some of them compared to the Gaussian policy in online and offline settings across several environments.

**Strengths:**

- To the best of my knowledge, there is no such study in the literature and I believe it is useful for RL in practice. Proposing more alternatives to the Gaussian policy for continuous state action space settings is interesting.
It would be nice to have the proposed parameterizations and their implementations integrated to existing RL packages to make them readily usable by practitioners.

- The execution of the study is solid: experiments are methodologically and rigorously conducted, figures are professional and the presentation is pleasant, this effort is much appreciated for a practical paper.

**Weaknesses:**

**Main Comments:**

- While the paper provides some insights, in terms of conclusions, it is hard to make a claim to extrapolate the findings beyond the interesting but expected fact that Gaussian policy might not always be the best choice and some other parameterizations might perform better. The paper does perform several experiments but  it is hard to derive actionable practical advice from 4 algorithms x 3 environments regarding which parameterization to be used in general.
I think the paper could do better to showcase the advantages and shortcomings of each one of the policies, especially to demonstrate the intuitive points mentioned in the introduction regarding each one of them. While the environments tested are standard benchmarks in RL, I think it would be interesting to design and/or consider simple environments with different levels of exploration needed for instance to show the differences between the policies. From the viewpoint of the practitioner, can we come up with a procedure to decide which one to use rather than myopically testing each one of them an decide accordingly? Can we design an index performance of exploration hardness to be exploited? Little can be said regarding this from the paper. Of course, we cannot reasonably be confident to choose the right parameterization since this would highly depend on the setting at hand as the paper shows but even heuristics would be useful. In other (older) ML areas, such procedures seem to be widely used in practice (e.g. for model selection in classification …)

- Other policy parameterizations than the Gaussian have been investigated in the literature as the paper acknowledges. Some additional relevant related work discussing the use of heavy tailed policies in RL and their benefits:

Amrit Singh Bedi and Anjaly Parayil and Junyu Zhang and Mengdi Wang and Alec Koppel. ‘On the Sample Complexity and Metastability of Heavy-tailed Policy Search in Continuous Control’, JMLR 2024.

S. Chakraborty et al., ‘Dealing with Sparse Rewards in Continuous Control Robotics via Heavy-Tailed Policy Optimization’, 2023 IEEE International Conference on Robotics and Automation.

- The paper could comment about the overhead computational effort (if any) required when going for more ‘sophisticated’ policy parameterizations than the simple Gaussian, notably in terms of sampling mechanism complexity and number of samples requirements, for instance depending on the dimensionality of the problem. Any differences regarding efficiency depending on the dimensionality of the problem?

- See questions below for clarifications.

**Minor comments:**
- l. 122-123: ‘In continuous action spaces, evaluating the log-partition function is generally intractable. Therefore, many researchers consider the Gaussian policy instead.’ I find this sentence a bit confusing. Is Gaussian policy only considered because BG policy cannot be used? In principle, one could consider any policy parameterization. For the continuous setting, it turns out that the Gaussian policy is the one of the simplest one can consider perhaps due to its omnipresence in statistics and parametric estimation as well as its widely available sampling procedure implementations.

**Questions:**

**Main questions (from the most to the least important):**

- Do all the policy neural networks have a similar number of parameters/weights across experiments for fairness of comparison?
- Are the implementations mostly readily available in existing RL packages or other libraries that could be directly be used? I see that you provide some of the sampling mechanisms but these seem to be already well known in the statistics literature given their popularity.
- Is GBMM only valid for 1 < q < 3?
- l. 41: ‘Heavy-tailed distributions could be more preferable as they are more robust’, what do you mean by robustness here? Can you elaborate more?
- Can you clarify the difference between the different Data settings in Fig. 8 (Medium-Expert, Medium, Medium-Replay)?

**Additional questions:**

- Any comments about the so-called Lévy alpha-stable distribution? It seems it has been explored in the literature to model the SGD noise.
See e.g. Simsekli et al. 2019. A Tail-Index Analysis of Stochastic Gradient Noise in Deep Neural Networks. ICML 2019.
- SAC encourages exploration. Would it be interesting to see if just changing the policy parameterization with a heavy tailed policy would already perform better than SAC? SAC seems to combine two effects: the parameterization of the policy itself and and the regularization to force the policy to be closer to a BG policy (why?)?
- More of a possible future work direction comment: This work explores in isolation each one of the parameterized policies to showcase the advantages and shortcomings of each one of them and compare them. Can we combine them to get the best of each one of them and have more flexibility like in ensemble methods or bagging approaches in Machine Learning? For instance we might need a very exploratory policy in some settings/regions of the action space or much less once we progress in policy optimization beyond the ability of single distributions to concentrate throughout learning.

**Details Of Ethics Concerns:**

Does not apply

---

> ### Author Response · Authors · 2024-11-20
> **we thank the reviewer's thoughtful comments**
>
> We sincerely appreciate the reviewer's thoughtful comments and appreciation of our work. *This comment answers the reviewer's points in the weakness section. The next comment will address the reviewer's questions*. All changes made to the paper were marked in blue.
>
> **Regarding behavior to hyperparameter changes**:
>
> We have included additional results in Figure 11, in the paper. It is a Manhattan plot visualizing the behaviour of SAC + all policies on all online environments. Although there is no definitive winner for all cases, we observe from Acrobot that the Student’s t and Gaussian have a similar behavior to hyperparameters changes. It may be safe to conclude that, if on a problem the Gaussian works, then the Student-t is very likely to work. And from Figure 5 (left), we know that Student-t is 75% more likely to outperform Gaussian given the same hyperparameter sweeping range. We have included this explanation to Section 5.1.
>
> **Regarding a procedure to pick policy**:\
> Several interesting future directions are possible for determining a suitable policy parametrization, including the model selection, such as the procedures the reviewer mentioned. Heuristically, the current study suggests that Gaussian and Beta are less preferred than heavy-tailed policies, especially Student’s t, which has shown stable improvements across environments, both in online and offline experiments. These improvements might be because the Student’s t maintains three learnable statistics: mean, standard deviation and degree of freedom (DOF). Learning the DOF allows the Student’s t to interpolate between the Gaussian and heavy-tailed distributions, granting it more flexibility. However, it remains an open question and interesting future direction to have a function that inputs the setting at hand and outputs a ranking on policy parametrizations with confidences (Student’s t > Heavy-tailed/Sparse q-Gaussian > Gaussian).
>
> **Regarding relevant references**:\
> We thank the reviewer for providing relevant references. We have incorporated them into the Introduction section.
>
> > With q > 1, we can obtain policies with heavier tails than the Gaussian, such as the Student’s t-distribution (Kobayashi, 2019) or the Levy Process distribution (Simsekli et al., 2019; Bedi et al., 2024). Heavy-tailed distributions could be more preferable as they are more robust (Lange et al., 1989), can facilitate exploration and help escape local optima in the sparse reward context (Chakraborty et al., 2023).
>
> **Regarding computational overhead**:\
> We have added the following paragraph regarding computation time to the last paragraph of Section 5.2:
>
> > To give an intuition for sampling time, we drew $10^5$ samples from a randomly initialized actor on two environments: HalfCheetah with 17-dim state and 6-dim action. The sparse $q$-Gaussian, heavy-tailed  $q$-Gaussian and Gaussian respectively cost (107.12, 72.09, 27.94) seconds. We confirmed that the sampling in Alg. 1 was on the same magnitude to the Gaussian, but the sparse $q$-Gaussian costed more than the heavy-tailed due to more computation to produce low-variance samples. This is further confirmed by  Hopper with 11-dim state, 3-dim action, where they cost (98.13, 65.17, 25.17) seconds.
>
> We expect the sampling time to be similar for common RL problems. But for the rare case where the dimension is huge, we expect that the efficiency for $q<1$ may be affected since it uses more computation to produce low-variance samples.  This is because for Gaussian, we first sample from $\mathcal{N}(0, I)$ and then transform it with the location and scale transform $\mu + \Sigma^{\frac{1}{2}}z$.
> In the case of GBMM, we sample from a uniform distribution followed by basic computation of cosine, q-log, and finally, the location-scale transform. Therefore, the computation overhead is similar to that of the Gaussian. For q<1, we used the stochastic representation. The computational cost slightly increases as we need to sample two variables and compute the radius and scaled matrix.
> By Alg.1, the computation overhead centers around computing the determinant $|\Sigma|$. For every update to the policy, we compute and store this determinant. If the matrix dimension is huge, then it may become the bottleneck. Fortunately, in our case, we can simply use the GBMM instead.
>
> **Regarding unclear statement of the Gaussian**:\
> We agree with the reviewer that our statement was not clear. We have removed this sentence and added the following explanation to the beginning of Section 3.1:
>
> > The Gaussian policy is one of the simplest distributions one can consider due to its omnipresence in statistics and parametric estimation as well as its widely available sampling procedure implementations. Since evaluating the log-partition function of BG is intractable, due to the aforementioned advantages many researchers consider the Gaussian policy instead.

---

> ### Author Response · Authors · 2024-11-20
> **addressing the reviewer's questions**
>
> We now address the reviewer's questions.
>
>  - **Q1**: Do all the policy neural networks have a similar number of parameters/weights across experiments for fairness of comparison?
>
> All policy networks share a common network architecture.  q-Gaussians and Gaussian are location-scale distributions; they output two distribution parameters: mean and standard deviation.
> Beta is slightly different, but the actor network also outputs alpha and beta shape parameters.
> Student’s t, by contrast, maintains three parameters: mean, deviation and the degree of freedom.
> In summary, all actor neural networks are similar for fair comparison; they output two statistics except for the Student’s, which outputs three.
>
> - **Q2**: Are the implementations mostly readily available in existing RL packages or other libraries that could be directly used? I see that you provide some of the sampling mechanisms but these seem to be already well known in the statistics literature given their popularity.
>
> The standard Gaussian/Squashed Gaussian and Beta policies are already available in the existing RL packages. However, we are unaware of RL libraries that provide q-Gaussians. We have implemented q-Gaussians and their sampling methods in PyTorch and will release the GitHub code base upon acceptance of the paper.
>
> - **Q3**: Is GBMM only valid for 1 < q < 3?
>
> GBMM can be used for all $q<3$. However, it tends to produce high variance samples, especially when $q$ is close to 1.  Therefore, for the sparse case ($q<1$), we resorted to the stochastic representation given by [Martins et al. 2022], which produces low variance samples only for sparse $q$-Gaussians.
>
>
> - **Q4:** l. 41: ‘Heavy-tailed distributions could be more preferable as they are more robust’, what do you mean by robustness here? Can you elaborate more?
>
> Heavy-tailed distributions like the Student’s t have been studied in [Kobayashi et al., 2019] to be resilient to environmental noises in that the policy is not easily affected by the noises so that it shifts to low-reward regions like the Gaussian. We have expanded our explanation in the corresponding paragraph to make it clearer.
>
> - **Q5**: Can you clarify the difference between the different Data settings in Fig. 8 (Medium-Expert, Medium, Medium-Replay)?
>
> We have added the following to Appendix D.3 for better clarification:
>
> > The D4RL offline datasets all contain 1 million samples generated by a partially trained SAC agent. The name reflects the level of the trained agent used to collect the transitions. The Medium dataset contains samples generated by a medium-level (trained halfway) SAC policy.
> Medium-expert mixes the trajectories from the Medium level and that produced by an expert agent. Medium-replay consists of samples in the replay buffer during training until the policy reaches the medium level of performance. In summary, the ranking of levels is Medium-expert > Medium > Medium-replay.

---

> > ### Author Response · Authors · 2024-11-20
> > **addressing the reviewer's minor questions**
> >
> > We now address the reviewer's minor questions.
> >
> >  - **Q1:** Any comments about the so-called Lévy alpha-stable distribution? It seems it has been explored in the literature to model the SGD noise. See e.g. Simsekli et al. 2019. A Tail-Index Analysis of Stochastic Gradient Noise in Deep Neural Networks. ICML 2019.
> >
> > The Levy Process distribution discussed in [Simsekli et al. 2019] seems similar to [Bedi et al., 2024, example 3]. The parameter is defined for $0<\alpha<2$ and it decays at the rate of $\frac{1}{|x|^{\alpha+1}}$.  If we would like to implement this distribution in RL, it may require more consideration since it has only $\alpha$-th moment when $\alpha < 2$. Typically, we would like to parametrize a distribution's mean and standard deviation. We thank the reviewer for providing the reference and have included it in our discussion.
> >
> >  - **Q2:** SAC encourages exploration. Would it be interesting to see if just changing the policy parameterization with a heavy tailed policy would already perform better than SAC? SAC seems to combine two effects: the parameterization of the policy itself and and the regularization to force the policy to be closer to a BG policy (why?)?
> >
> > Indeed, SAC can be seen as combining the two effects. However, analyzing how it facilitates exploration and boosts performance can be tricky because the entropy is computed by an empirical average of log-likelihood, which is tightly related to the distribution shape and policy parametrization. Second, it forces the policy to be close to BG, but it is hard to know which policy parametrization would obtain a smaller projection error. In Figure 4 Mountain-car, we see that Beta is the best with SAC, while Gaussian performs similarly to other policies. Given that Beta is the only non-location-scale family member, it seems difficult to decouple the contribution of SAC: it could be because Beta is more flexible, so it could approximate BG better, or it could also be because Beta had higher entropy than the others. Carefully studying these topics is an interesting future direction.
> >
> >  - **Q3:** More of a possible future work direction comment: This work explores in isolation each one of the parameterized policies to showcase the advantages and shortcomings of each one of them and compare them. Can we combine them to get the best of each one of them and have more flexibility like in ensemble methods or bagging approaches in Machine Learning? For instance we might need a very exploratory policy in some settings/regions of the action space or much less once we progress in policy optimization beyond the ability of single distributions to concentrate throughout learning.
> >
> > We thank the reviewer for pointing out an interesting future direction. It is indeed possible. Since the $q$-Gaussian family covers all other distributions (except Beta) by varying $q$. The suggested idea may potentially be achieved by maintaining an ensemble of $q$-Gaussians each with different $q$ to have more flexibility.

---

> > > ### Comment · Reviewer_dEFf · 2024-11-30
> > > **Thank you for your response**
> > >
> > > I would like to thank the authors for their clear responses that answer my questions. I am increasing my score. The presentation is clear, the claims are aligned with the evidence provided, extensive experiments are conducted. Most importantly, for a work focused on practice, I maintain that the rigour in conducting the experiments and presenting the results (faithfully) is much appreciated and not often seen at this level of execution from my experience. The appendix contains many details and additional experiments that give further confidence about the reproducibility of the results.
> > >
> > > The question addressed in this work is not very original in my opinion as one might expect that diverse policy parameterizations certainly need to be considered for continuous state action space settings and may have different benefits. Nevertheless, to the best of my knowledge, it has not been rigorously investigated in practice (although a few works already alluded to that). I believe this is important for practical RL and this work does a step in that direction. I think much more is to be done to provide more value to practitioners in this regard (see e.g. some questions I mentioned above which are partially answered in the paper in special settings without systematic heuristic proposals that can be readily used by practitioners for instance), the paper has some limitations from this viewpoint in my opinion but this might also open avenues for future work. I keep supporting acceptance of this paper.

---

### Official Review · Reviewer_nnsC · 2024-11-04

**Soundness:** 3
**Presentation:** 3
**Contribution:** 2
**Rating:** 6
**Confidence:** 3

**Summary:**

This paper focuses on solving reinforcement learning problems with continuous action spaces using policy optimization methods. In such cases, a typical solution for handling continuous action spaces is to impose a probability distribution (specified by finitely many parameters) on the continuous space. A common choice is the Gaussian distribution. The authors propose using a different family of probability distributions: the q-exponential family. Extensive numerical simulations are provided.

**Strengths:**

The motivation for using the q-exponential family of distributions to encourage exploration seems intuitive. The numerical simulations are quite extensive and successfully demonstrate that it is not always optimal to use a Gaussian distribution to parameterize the policy in reinforcement learning.

**Weaknesses:**

I suspect that there is an exploration vs. exploitation trade-off in using heavy-tailed probability distributions. Is it possible to design a synthetic example to verify this? Additionally, once this is verified, there might be a way to switch probability distributions during training to balance the exploration-exploitation trade-off, which seems like an interesting direction to investigate.

**Questions:**

The results are clear and I do not have questions.

---

> ### Author Response · Authors · 2024-11-20
> **added additional result to help conclude Student's t as the policy to go**
>
> We thank the reviewer for the helpful feedback and pointing out an interesting and important future direction.
>
> Indeed, there may be an exploration vs. exploitation trade-off underlying the $q$ policies. In the paper, this role was played by the Student's t whose degree of freedom was also learned, so it could switch between a Gaussian (so it could shrink into a Delta-like policy) and a heavy-tailed distribution. It is an interesting future direction to see what trade-off each policy parametrization would make. For general $q$-Gaussians, this could potentially be achieved by adaptively adjusting the $q$ parameter.
>
> To further consolidate the paper and provide intuition to the question of **"what policy should we use?"**, we have included additional results in Figure 11, a Manhattan plot to visualize SAC’s behaviour to hyperparameters changes. Although there is no conclusive winner, we can safely conclude from the plot of Acrobot that if we have a problem where the Gaussian works, the Student’s t is very likely to work, and from Figure 5(left), it is 75% more likely perform better given the similar hyperparameter sweeping behaviour. Combined with the existing offline results, these observations suggest that the researchers can use Student’s t as a drop-in replacement for the Gaussian and benefit from the performance improvements.

---

### Author Response · Authors · 2024-11-24

Dear Reviewers,

We would like to thank you again for your time and thoughtful feedback on our paper. As the discussion phase approaches its deadline, we would like to kindly ask if there are any additional comments or questions you might have. We value the opportunity to interact with you as you insights have been helpful in improving our paper. Thank you for your support, and we look forward to hearing from you.

Best regards,\
The Authors

---

### Meta-Review · Area_Chair_kERP · 2024-12-23

**Metareview:**

This paper proposes a general class of policy optimization algorithms for continuous action spaces which replaces the Gaussian by general q-exponential family. This paper provides a discussion on the choice of q value and their effectiveness, and conclude on the general advantage of using student t-distribution as policy. The problem is important but lack study in the field. All reviewers are convinced by the importance of such study and by the contribution made by this paper to the field. We thus recommend acceptance.

**Additional Comments On Reviewer Discussion:**

All reviewers are in favor of accepting this paper.

---

### Decision · Program_Chairs · 2025-01-22

Accept (Poster)